# The Climate of the Common Era off the Iberian Peninsula

Abrantes[1,2], Fátima; Teresa Rodrigues[1,2], Marta Rufino[2,3]; Emília Salgueiro[1,2]; Dulce Oliveira[1,2,4]; Sandra Gomes[1]; Paulo Oliveira[1]; Ana Costa[5]; Mário Mil-Homens[1]; Teresa Drago[1, 6]; Filipa Naughton[1,2,]

1 - Portuguese Institute for Sea and Atmosphere (IPMA), Divisão de Geologia Marinha (DivGM), Rua Alferedo Magalhães Ramalho 6, Lisboa, Portugal

2 - CCMAR, Centro de Ciências do Mar, Universidade do Algarve, Campus de Gambelas, 8005-139 Faro, Portugal

3 - IFREMER - Centre Atlantique (French Research Institute for Exploitation of the Sea, Département Ecologie et Modèles pour l'Halieutique (EMH), Rue de l'Ile d'Yeu - BP 21105, 44311 Nantes cedex 3, France

4 - Université de Bordeaux, EPOC, UMR 5805, F-33615 Pessac, France

5 - Centro de Investigação em Biodiversidade e Recursos Genéticos (EnvArchCIBIO/InBIO) and Archaeosciences Laboratory (LARC/DGPC), Rua da Bica do Marquês, 2, 1300-087, Lisboa, Portugal

6 - Instituto Dom Luiz, Universidade de Lisboa, 1749-016 Lisboa, Portugal

*Correspondence to*: Fatima Abrantes (Fatima.abrantes@ipma.pt)

Key Words – Last 2,000 yr, climate, SST, precipitation, Iberian Peninsula

**Abstract.** The Mediterranean region is a climate hot spot, sensitive not only to global warming but also to water availability. In this work we document major temperature and precipitation changes in the Iberian Peninsula and margin during the last 2,000 yr, and propose an interplay of the North Atlantic internal variability with the three atmospheric circulation modes (ACM), (North Atlantic Oscillation (NAO), East Atlantic (EA) and Scandinavia (SCAND)) to explain the detected climate variability.

We present reconstructions of Sea Surface Temperature (SST derived from alkenones) and on-land precipitation (estimated from higher plant n-alkanes and pollen data) in sedimentary sequences recovered along the Iberian Margin between the South of Portugal (Algarve) and the Northwest of Spain (Galiza) (36 to 42 ºN).

A clear long-term cooling trend, from 0 CE to the beginning of the 20th century, emerges in all SST records and is considered to be a reflection of the decrease in the Northern Hemisphere summer insolation that began after the Holocene optimum. Multi-decadal/ centennial SST variability follows other records from Spain, Europe and the Northern Hemisphere. Warm SSTs throughout the first 1300 yr encompass the Roman Period (RP), the Dark Ages (DA) and the Medieval Climate Anomaly (MCA). A cooling initiated at 1300 CE, leads to 4 centuries of colder SSTs contemporary with the Little Ice Age (LIA) while a climate warming at 1800 CE marks the beginning of the Modern/Industrial Era.

Novel results include two distinct phases in the MCA, an early period (900 – 1100 yr) characterized by intense precipitation/ flooding and warm winters but a cooler spring-fall season attributed to the interplay of internal oceanic variability with a positive phase in the three modes of atmospheric circulation (NAO, EA and SCAND). The late MCA is marked by cooler and relatively drier winters and a warmer spring-fall season consistent with a shift to a negative mode of the SCAND.

The Industrial Era reveals a clear difference between the NW Iberia and the Algarve records. While off NW Iberia variability is low, the Algarve shows large amplitude decadal variability with an inverse relationship between SST and river input. Such conditions suggest a shift in the EA mode, from negative between 1900 and 1970 CE to positive after 1970, while NAO and SCAND remain in a positive phase. The particularly noticeable rise in SST at the Algarve site by the mid 20th century (± 1970), provides evidence for a regional response to the ongoing climate warming. The reported findings have implications for decadal-scale predictions of future climate change in the Iberian Peninsula.

## 1 Introduction

Today's anthropogenically-induced global warming poses a pressing problem on societies' sustainability (IPCC, 2013a, b). Regions such as the Arctic and the Mediterranean are highlighted as the most sensitive and potentially vulnerable to ongoing global warming (Climate, 2011; Giorgi, 2006).

Global and regional model simulations for the Iberian Peninsula forecast temperatures rising above the predicted global mean, and changes in precipitation consistent with long dry summers and a short and wetter rainy season particularly for the southern region (Guiot and Cramer, 2016; Miranda et al., 2002). Most of this knowledge is based on the analysis of instrumental data and modeling of global and hemispheric average conditions. However, given the limited time-span covered by the instrumental data, and to better comprehend the impact of climate warming it is essential to analyze and understand the response of the climate system in perspective of a longer time-scale. In light of the current warming, previous warm periods and warming transitions such as those occurring over the last 2,000 yr are of particular interest. Proxy-based climate reconstructions and modeling of the climate for the last 1 or 2 millennia in the Northern Hemisphere, have identified the late Roman Period (RP 0-500 CE), the Dark Ages (DA; 500-900 CE) and the Medieval Climate Anomaly (MCA; 900-1300 CE), also known as Medieval Warm Period (MWP), and the Little Ice Age (LIA; 1350-1850 CE), and attribute these climate variability mainly to external forcing such as solar and volcanic activity (Fernandez-Donado et al., 2013; Hegerl et al., 2006; Jones et al., 2001; McKim, 1998; Schurer et al., 2014). After 1900 CE, the rise in global mean atmospheric temperature is mainly attributed to the unprecedented increase of greenhouse gases in the atmosphere (IPCC, 2013a).

Throughout the last two decades, many high resolution 2,000 yr climate records have been generated from Iberian lake sediments (e.g. Hernández et al., 2015; Jambrina-Enríquez et al., 2016; Morellón et al., 2009; Moreno et al., 2008; Valero-Garcés et al., 2006); speleothems (e.g. Martín-Chivelet et al., 2011) and marine sediments (e.g. Abrantes et al., 2005; Abrantes et al., 2011; Desprat et al., 2003; Diz et al., 2002; Lebreiro et al., 2006; Pena et al., 2010). Those individual as well as the compiled evaluations of climate evolution over Iberia (e.g. (Moreno et al., 2011; Sánchez-López et al., 2016) reveal multi-decadal to centennial climate variability in accordance with the main patterns identified for the North Hemisphere (Ahmed et al., 2013; Büntgen et al., 2011; Cook et al., 2004; Esper et al., 2002; Luterbacher et al., 2016; Moberg et al., 2005 ).

Additionally, given the dominance of the North Atlantic Oscillation (NAO) (Hurrell, 1995) in the Northern Hemisphere, most of the above works attribute the variability to changes in the prevailing modes of the NAO (Abrantes et al., 2005, 2011; Lebreiro et al., 2006). In recent years it has been proposed that the East Atlantic (EA) and Scandinavia (SCAND) modes also play a significant role on North Atlantic climate evolution (Comas-Bru and McDermott, 2014; Hernández et al., 2015; Jerez and Trigo, 2013). Sánchez-Lopez et al., (2016), on the basis of a spatiotemporal integration of several climate reconstructions, attempted to identify the role of those atmospheric patterns over the Iberian Peninsula. Their results reveal E-W and N-S humidity gradients from 0 to 500 CE and between 500 and 900 CE respectively, while between 900 and 1850 CE temperature and humidity conditions are more homogenous throughout the Peninsula. These conclusions support atmospheric pathways as the main control of climate variability in Western Europe on multi-decadal time-scales. However, Yamamoto and Palter (2016) observed a clear relationship between the Atlantic Multidecadal Oscillation (AMO) and the atmospheric circulation over Europe, with northerly winds associated to a positive state of the AMO and zonal winds to a negative state of the AMO. To better understand the role of oceanic and atmospheric processes on past climate and their relevance to the Iberian Peninsula's future climate, it is pivotal to obtain more high-resolution climate archives of the latest centuries and millennia. Here, we explore the main oceanic and atmospheric processes that drive complex spatial climate patterns over the Iberia Peninsula across the last 2,000 yr by integrating new records from the Iberian margin, Galiza, Minho and Algarve, with published datasets (Porto and Tejo; Abrantes et al., 2005, 2011; Lebreiro et al., 2006).

**2 Oceanographic Conditions**

The western coast of the Iberian Peninsula (Fig. 1) is characterized by contrasting oceanographic conditions between a spring-summer (April to October) coastal upwelling regime and a surface equatorward current (Fiuza, 1982, 1983; Peliz et al., 2002; Relvas et al., 2007), and a winter alongshore poleward warm counter current (Fiuza and Frouin, 1986; Peliz et al., 2005).

The spring-summer upwelling, constitutes the northern part of the Eastern North Atlantic Upwelling / Canary System and is connected to the presence of the Azores high-pressure system and the development of northerly alongshore winds (Fiuza, 1983). Upwelled waters are transported southwards by a jet-like surface current (Fig. 1C), the Portuguese Coastal Current, which is the coastal component of the Portuguese Current that branches of the North Atlantic Current, (Fiuza, 1982, 1983; Fiúza and Macedo, 1982). On the southern coast (Algarve), upwelling favorable conditions are rare, but western upwelled waters flow around Cape S. Vicente and along the south coast, (Fiuza, 1982, 1983; Fiúza and Macedo, 1982; Sánchez and Relvas, 2003) and can spread to its easternmost sector (Cardeira et al., 2013). This eastward flow of cold western upwelled waters alternates with the propagation of westward flows related to warm water and increased vertical stratification showing a direct relationship between flow velocity and water temperature (Garel et al., 2016; Relvas and Barton, 2002).

In winter, the prevalence of westerly/southwesterly winds leads to the intensification of the Iberian Poleward Current (Fig. 1B). This current, which is a branch of the Azores Current, consists of an upper slope/shelf break poleward flow that transports saltier and warmer (subtropical) waters (Peliz et al., 2005) depending mostly on the intensity of the southerly winds (Teles-Machado et al., 2015). Another important feature of the winter circulation over the western margin is the formation of coastal buoyant plumes, characterized by low salinities and temperature lower than the ambient shelf waters (Peliz et al., 2005). Such plumes result from the freshwater discharge from rivers, thus reflecting continental precipitation. Precipitation occurs mainly in winter, as a result of the moisture carried by the westerly winds into the Peninsula, and has important latitudinal differences, from 500 mm/year in the southeast to >3000 mm/year in the northwestern area (Miranda et al., 2002). As a consequence, buoyant plumes are mainly associated to the major northern Portuguese rivers (Minho, Douro, Mondego) but are also linked to the Tagus River, and can either develop into inshore currents (under typical winter downwelling conditions), or spread offshore under northerly wind conditions (Iglesias et al., 2014; Marta-Almeida et al., 2002; Mendes et al., 2016; Oliveira et al., 2007; Otero et al., 2008).

**3 Material and Methods**

This study combines proxy data previously published for sedimentary sequences collected off the Tagus River (PO287-26B, 26G, D13902 and D13882 designated as Tejo; Abrantes et al., 2005; Rodrigues et al., 2009) and Douro River (PO287-6B, 6G, designated as Porto; Abrantes et al., 2011) with new data from 3 other sites on the Iberian Margin (Table 1, Fig. 1). Two of the new sites are located in the northern area, off Vigo (GeoB11033-1 referred to as Galiza), and off the Minho River mouth (Diva09 GC, referred as Minho in this paper), and one core from southern Iberian/Algarve margin (POPEI VC2B, referred as Algarve). With the exception of the Galiza deep-sea core, all other sedimentary sequences were collected in the inner shelf in areas directly affected by river discharge.

Age-models of the three new cores (Galiza, Minho and Algarve), were constructed using the methods of Abrantes et al., (2005, 2011) and are based on accelerator mass spectrometry radiocarbon (AMS [14]C) and [210]Pb-inferred dates (section 1, Table SM1, Figs. SM1 and SM2 in Supplementary Material). Raw AMS [14]C dates were corrected for marine reservoir ages of 400 yr ±36 (Abrantes et al., 2005) and converted to calendar ages using INTCAL04 (Reimer et al., 2004). The obtained calendar ages are presented in years Anno Domini, now designated by Common Era (CE - McKim, 1998).

Sea Surface Temperature (SST) was determined based on the ratio of alkenones ($U^{k'}_{37}$) synthesized by coccolithophores (phytoplankton group) (e.g. Eglinton et al., 1992; Rosell-Melé et al., 1994; Villanueva and Grimalt, 1997). Lipid compounds synthesized by higher plants, such as C23–C33 n-alkanes ([n-alk]) (e.g. Eglinton and Hamilton, 1967) and the total pollen concentration (TPC) were used as indicators for river discharge intensity and on-land precipitation regime (e.g. Rodrigues et

al., 2009). Major changes in vegetation cover, that is, continental temperature and moisture conditions, were evaluated from pollen assemblages (Naughton et al., 2007).

Alkenones and higher plant n-alkanes were analyzed on 2 g of homogenized sediment using a Varian gas chromatograph Model 3800 equipped with a septum programmable injector and a flame ionization detector at the DivGM-IPMA laboratory according to the methods described in Villanueva (1996) and Villanueva et al. (1997). Analytical error was 0.5°C. The concentration of each compound was determined using hexatriacontane as internal standard. For the calculation of Sea Surface Temperature (SST) we selected the globally defined calibration of Muller et al. (1998) ($U^{k'}_{37}$ = 0.033xSST-0.044). Both alkenone based SST and [n-alk] are widely accepted proxies for SST and river input, but to better understand our regional records, the significance of our high-resolution sediment data was assessed by comparison to NOAA daily Optimum Interpolation Sea Surface Temperature (OISST, V2 AVHRR-only) dataset (Fig. SM3, Supplementary Material), and the river discharge dataset at *Sistema Nacional de Informação de Recursos Hídricos* (SNIRH).

Sample preparation procedures for pollen analyses followed the methods described in Naughton et al. (2007). Pollen and spores were counted using a Nikon light microscope at x550 and x1250 (oil immersion) magnification. Pollen identification was done via comparison with the pollen atlases of Moore et al. (1991) and Reille (1992). A minimum of 100 *Lycopodium* grains, 20 pollen types and 100 pollen grains, excluding the over represented *Pinus,* have been counted (Naughton et al., 2007). Pollen was gathered in two main groups: AP (arboreal pollen) including all trees and shrubs but excluding the over represented pine taxa, and the semi-desert plants, which groups xerophytic shrubs of semi-desert habitats (*Artemisia*, Chenopodiaceae, *Ephedra*).

To reduce the local variability signal and better evaluate the multi-decadal record at the regional level, a stack off all the cores was generated from the SST and n-alkanes original records on their original age-models. Each record was standardized (subtracting each value by the mean) and scaled (dividing the centered columns by its standard deviation). This technique weights high-resolution records more heavily and prevents interpolation across gaps or hiatuses from affecting the stack (Lisiecki and Raymo, 2005). Additionally, a test to the sensitivity of the stack to the bin-size and the inadequacies of the proxy data (Supplementary Material) reveals that the stack is insensitive to bin sizes (20 – 50 yr) and independent of data deficiencies (Fig. SM4). To focus on lower variability periods a similar stack was built using a 5-yr bin size and the Porto, Tejo and Algarve cores with sedimentation rate > 0.4 cm yr⁻¹ (Fig. SM, 3rd panel).

To investigate the existence of periodic signals and potential changes in their amplitude through time, we carried out a continuous time series transformation with a Morlet wavelet analysis on each dataset and stack (Torrence and Compo, 1998), after interpolation of the series to regular time steps. Interpolation was done using a cubic splines method and the temporal resolution of the interpolation was established as half of the absolute median difference between every consecutive time interval. Data was then detrended using a modified negative exponential curve, as required for the analysis. All statistical analysis was done using the libraries biwavelet (Akima and Gebhardt, 2016; Gouhier et al., 2016), from r-project (R Core Team, 2016).

Primary data for the new sedimentary sequences is archived in PANGEA (doi:10.1594/PANGAEA.882269).

## 4 Results

All the sedimentary sequences selected for this study have exceptionally high sedimentation rates in the upper levels, allowing for very high temporal resolution of 2-3 yr in the top sediments. In the older part of the sequences, larger sampling intervals and/or lower sedimentation rates provide a temporal resolution of ±30 yr. Additionally, the analyses of SST and multiple proxies for on-land precipitation from the same sediments, allows us to accurately evaluate the coupled ocean and land variability without chronological ambiguity. Furthermore, to better assess the regional value of Uk'₃₇-SST and [n-alk], sediment derived variables were compared to climatological datasets (sections 2 and 3 in Supplementary Material). Results

validate the proxies and reveal that in the west coast SST compares to winter climatological data but in the Algarve SST mimics spring-fall (Fig. SM2), providing the opportunity to disentangle winter from spring-fall conditions in the region.

## 4.1 Sea Surface Temperature (SST)

SSTs are minima off Porto (14 to 16 ºC), maxima in the Algarve (17 to 20 ºC), while average temperatures are observed in the Tejo, Minho and Galiza (15 to 18 ºC) as shown in Fig. 2A by the alkenone-derived SST reconstructions for all 7 Iberian margin records. The temperature difference between the areas is maintained throughout the last 2,000 yr but the amplitude of decadal-secular variability is higher at the Tejo and Algarve sites (3ºC) than the 1.5 ºC detected in the northern sites (Porto, Minho and Galiza) (Fig. 2A). Moreover, the SST stack (Fig. 2B), reveals an overall long-term cooling trend from 0 CE to the beginning of the 20[th] century, with a stronger gradient at the Tejo (2.5 ºC/ 2,000 yr) than at the other sites (1ºC/ 2,000 yr) (Fig. 2A). This long-term cooling follows the gradual decrease of northern Hemisphere summer insolation across the Holocene reported for this region by Rodrigues et al. (2009), but also observed in other European and worldwide records (Ahmed et al., 2013; Luterbacher et al., 2016; McGregor et al., 2015).

Superimposed on the long-term cooling, both the individual SST records (Fig. 2A) and the SST stack (Fig. 2B) display century-scale variability comparable with that recorded in both the oceanic and continental environments of northern Spain (Fig. 2C, D) as well as in other European and Northern Hemisphere records (Fig. 2E, F) (Diz et al., 2002; Luterbacher et al., 2016; Martín-Chivelet et al., 2011; Moberg et al., 2005). Relatively high SSTs occur during the first 9 centuries encompassing the late Roman period (RP; 0 – 500 CE) and Dark Ages (DA; 500 -900) (Fig. 2A, B), mainly at the Southern sites (Tejo and Algarve; Table 2). Consistently warmth conditions are also recognized at all sites between 900 and 1300 CE within the MCA (Fig. 2A, B). The Western Iberia records reveal a warmer first phase in the MCA (900 – 1150 CE) in agreement with data for thee NE North Atlantic (Cunningham et al., 2013). However, at the Algarve site SST decreases during the MCA, between 800 and 1100 CE followed by an increase of ± 1 ºC between 1100 and 1300 CE.

A transition from warm to cold climatic conditions starts around 1300 CE (Fig. 2B) associated with the Wolf solar minimum (Fig. 2H) (Bard et al., 2007). Cold conditions prevail for most of the 15[th] to 18[th] centuries in Western Iberia, during the well-known LIA (Bradley and Jones, 1993). SSTs are colder than during MCA by an average 0.5 ºC in the northern sites and 1.2ºC in the southern sites, particularly in the high resolution LIA record from the Algarve (Fig. 2A). Abrupt cold episodes are synchronous with solar minima and major volcanic events (Fig. 2G, H) (e.g. Crowley and Unterman, 2013; Solanki et al., 2004; Steinhilber et al., 2012; Turner et al., 2016; Usoskin et al., 2011).

At 1800 CE an increase in SST marks the transition to warm modern times referred to as the Industrial Era. During the 20[th] century, unusually large decadal-scale SST oscillations are recorded in the southern sites, in particular in the Algarve, where a more abrupt rise in SST occurs by the mid 20[th] century at around 1970 CE (Fig. 2A) coinciding with the Great Solar Maximum (1940 – 2000) (Usoskin et al., 2011).

Such increase in the amplitude of SST variation during the last 50 years, in particular at the Algarve site, although attributable to better proxy preservation in the more recent sediments (Calvert and Pedersen, 2007), is certainly also a reflection of the regional reaction to an intensification of climatic extremes, an expected response to the ongoing climate warming (IPCC, 2013a; Miranda et al., 2002).

## 4.2 Continental Precipitation

At present there is a clear north-south difference in the precipitation regime, with higher mean annual precipitation in the north versus the southern region (Miranda et al., 2002). In our records, the lowest lipid compounds synthesized by continental plants ([n-alk]) (100-700 ng/g) are found at Galiza, the deepest (1873 m) and more oceanic site, while the highest [n-alk] (1000 to 7000 ng/g) mainly reflect the large Douro River discharge. The Minho, Tejo and Algarve sites that are

influenced by intermediate mean annual discharge rivers Minho, Tagus and Guadiana (Miranda et al., 2002), show intermediate [n-alk] (700 - 4000 ng/g) (Fig. 3A).

The total [n-alk] stack (Fig. 3B) highlights abrupt decadal-scale variability of river discharge throughout the first 900 yr, with relatively lower mean values during the DA (500 to 900 CE) (Fig.3B). In the MCA, a strong positive deviation occurs in the early MCA (980-1100 CE), mainly reflecting the extreme river discharge in Porto, since reduced river discharge is recorded at the Tejo and Algarve sites (Fig. 3A). From 1100 to 1200 CE river discharge is still low in the southern cores and Porto reaches minimum values. After 1200 CE, throughout the late MCA to the beginning of the LIA, a gradual increase in the total [n-alk] reflects a persistent river flow at all latitudes (Fig. 3A, B). As found for SST, river input during the Industrial Era shows large amplitude variability in the southern cores and the Algarve in particular.

Iberian precipitation variability reflects changes in river discharge intensity, which can also be estimated from oscillations in TPC (Naughton et al., 2009). TPC data, although at a very low temporal resolution, is available for 3 locations, Minho, Tejo and the Algarve (Fig. 3C). The TPC data suggests a much larger river discharge during the RP at the Tejo (core D13882) than in the Minho or the Algarve. During the DA and the MCA TPC-derived riverine input/precipitation is higher at the Minho and Tagus hydrological basins than in the Algarve, but at 1400 CE the Algarve record rises to higher values comparable to those observed at the Minho, in accordance with [n-alk] (Fig. 3A, C).

To substantiate our [n-alk] and TPC records of on-land precipitation, we compared our data to various reconstructions and historical documents of Iberian flood events. For the Tagus, flood reconstructions based on the hydrological basin terraces (Benito et al., 2005; 2004; 2003) and the Taravilla Lake sediment record from the headwaters of the Tagus River (Moreno et al., 2008). Historical documentation was provided by Tullot (1988) for the Douro and Minho Rivers, and by Barriendos and Rodrigo (2006) for most Iberian Peninsula basins including the coastal Mediterranean and daily journals for the most recent flood events of the Guadiana River (Barriendos and Martin-Vide, 1998; Barriendos and Rodrigo, 2006; Cabrita, 2007; Varzeano, 1976) (Table 3; Fig. 3). Despite age uncertainties, the stack [n-alk] maxima between 980 and 1100 CE coincides with reports of major flooding events in both the Douro and Minho Rivers (Tullot, 1988). Other periods marked by strong precipitation occur in the MCA between 1180 – 1200 CE, and again in the beginning of LIA, from 1450 to 1470 CE (Tullot, 1988). At the Tejo site the [n-alk] record also agrees with significant river flooding phases (1200 – 1300 CE; 1950 – 1980 CE) (Benito et al., 2004; Benito and Hudson, 2010; Benito et al., 2003). The Algarve site, located 80 km to the west of the Guadiana River mouth appears to be recording not only the most recent newspaper's reported flooding events of 1876 and 1979 CE (Cabrita, 2007; Varzeano, 1976), but also the Atlantic basin flooding events (Barriendos and Rodrigo, 2006; Benito et al., 2004). The similarity of the independently identified records of storm/flooding periods for the various regions, leads to the conclusion that the maxima in [n-alk] can indeed be attributed to extreme precipitation and flooding conditions (Fig. 3).

### 4.3 Atmospheric Temperature

Arboreal (AP) and semi-desert pollen variability at the Minho, Tejo and Algarve sites were compared with the same pollen curves from the Ria de Vigo (Desprat et al., 2003) and were found to reflect the main forest/climate changes over the last 2,000 yr (Fig. 4A, B). Major forest expansion, revealed by increasing AP percentages, occurs during the RP and the MCA, indicating relatively warm conditions on-land (Fig. 4B). In contrast, a reduction of the forest cover, revealed by the decrease of arboreal pollen, suggests relatively cold conditions during the LIA (Fig. 4B). Between 1700-1800 CE there is a strong decrease of arboreal pollen in Algarve and Minho suggesting an abrupt cooling episode over Iberia (Fig. 4B). After 1800 CE a new increase in AP reflects climate warming over the continent. AP variability in western Iberia over the last 2,000 yr shows no clear match with European summer atmospheric temperature or spring precipitation (Fig. 4F, G), but agrees with the general trend of Northern Spain's stalagmite T-anomaly (Fig. 4E) (Martín-Chivelet et al., 2011) and AP variability at the Ria de Vigo (Fig. 4B) (Desprat et al., 2003).

Semi-desert plant values are highest at the Algarve (although with a modest 8% contribution), lower at the Tejo site and further reduced at the Minho, showing clearly the north to south precipitation contrast (Fig. 4A). Between 1700 and 1800 CE (within the LIA), a marked decrease of both arboreal taxa and semi-desert plants is observed and co-occurs with abrupt shifts between periods of increased frequency of strong rain events (Barriendos and Martin-Vide, 1998) and periods of prolonged drought (Barriendos, 2002; Benito and Hudson, 2010). Furthermore, this interval also coincides with a period of stable temperature in Northern Spain, and minima in European seasonality and spring atmospheric temperature in Europe (Fig. 4E, H, I) (Luterbacher et al., 2004) as well as a minimum in spring-summer and winter precipitation (Fig. 4C, J, K) (Martín-Chivelet et al., 2011; Romero-Viana et al., 2011; Touchan et al., 2005).

## 5. Discussion

### 5.1 Climate Forcing Mechanisms

Climate reconstructions for the Iberian Peninsula are complex because the seasonal variation of the oceanographic system along the Iberian Peninsula margin generates multiple conditions which combination is likely to vary through time. Nevertheless, the use of multiple proxies for each core, and at several sites, together with the regional anomaly stack, allow for the more robust climatic configurations to be identified. Furthermore, our assessment of the regional SST records (Supplementary Material, Fig. SM2) indicates that SSTs are comparable with winter temperatures in the west coast while in the Algarve SST match spring-fall temperatures (Fig. SM2), giving us the opportunity to disentangle winter from spring-fall conditions in the region.

The Iberian Peninsula margin was relatively warm from 0-1300 CE, in particular during the final stage of the RP (0- 500 CE) and the MCA (900-1300 CE), while SST slightly decreases during the DA (500 -900 CE) (Fig. 2B). A clearly colder LIA lasts from 1350 to 1850 CE, when a rise in SST marks the transition to modern times / Industrial Era. This initial augment in SST is followed by a second more abrupt mid 20[th] century SST rise, particularly evident in the southernmost site (Fig. 2A).

In terms of on-land precipitation, the early MCA is a period of extreme precipitation and flooding, mainly from the Douro River (Porto site) (Fig 3A, B). During the LIA, frequent but less extreme precipitation, is inferred at all sites with apparent periods of flooding that are in agreement with other flooding records (e.g. Barriendos and Martin-Vide, 1998; Benito et al., 2005; Cabrita, 2007; Moreno et al., 2008; Tullot, 1988; Varzeano, 1976). Southern Iberian margin records also show evidence of the well-known major storm events of the 20[th] century.

The dominant large-scale climate mode operating in the Northern Hemisphere is the North Atlantic Oscillation (NAO) (Hurrell, 1995). Being mainly a winter season mode that varies on scales of days to decades, it translates into strong northerly winds and coastal upwelling favorable conditions during positive phases, while during NAO negative phases (also known as "blocked"), westerly/southwesterly winds predominate and result in very cold winters and increased storm activity (Hurrell, 1995; Trigo et al., 2004).

Other prominent atmospheric circulation modes, the EA and the SCAND (Comas-Bru and McDermott, 2014; Jerez and Trigo, 2013), constitute second leading modes that interplay with the NAO. Their temporal variability must have also played a role on climatic evolution in the North Hemisphere. The EA has a strong influence on the strength and location of the NAO dipoles mainly on multi-decadal time-scale, and exerts major control on winter and summer temperature over the Iberian Peninsula (Hernández et al., 2015). The SCAND functions as a blocking high-pressure system that changes the westerly winds path and influences southwestern Europe mainly during its positive phase, when it contributes to below average temperatures and above average precipitation (Hernández et al., 2015). The regional effect of all three ACMs on SST and precipitation over the Iberian Peninsula, for winter and summer periods, is presented in figure 5 of Hernández et al. (2015). Bearing in mind the Atlantic coast, besides the negative relationship observed between NAO and winter precipitation, the EA and SCAND are also positively related to winter precipitation, particularly on the Northern Iberian Peninsula. In

summer, there is a slight positive relationship between SCAND and precipitation in the north but an important negative effect with temperature in most of the south.

The compilation of Sánchez-Lopes et al, (2015), concludes that the climate of the Iberian Peninsula during the last 2,000 yr has been modulated by the combined effect of two main modes of atmospheric circulation, the NAO and the EA. Negative NAO and positive EA generate warm atmospheric temperatures and higher humidity in the west and south of the Iberian Peninsula during the RP. NAO positive and EA negative cause a more humid north with a W-E humidity gradient during the DA. Consistently warm and dry conditions during the MCA throughout the Iberian Peninsula are attributed to NAO positive and EA positive modes. On the contrary, NAO and EA negative modes are considered to explain the cold and wet winters, as well as, the cold summers proposed for the LIA.

Although our results might be explained by the above-described mechanisms proposed Sánchez-Lopez for the RP, the DA and the LIA (Figs. 2,3,4), the climate conditions detected in the MCA and the Industrial Era, need further discussion.

## 5.2 The particular case of the MCA and the industrial era

### 5.2.1 The MCA: precipitation distribution and the storm track

Dry and warm winters as well as warm summers are likely to be generated by the prevalence of NAO and EA positive modes as proposed by Sánchez-Lopes et al., (2015). Stronger coastal upwelling conditions have also been suggested to explain the productivity record of the Tejo site. This would imply prevailing northerly winds and an active Portuguese Current, which would equate to a positive NAO-like state or frequent occurrence of extreme NAO maxima during the MCA (Abrantes et al, 2005), that agrees with the NAO reconstruction of Ortega et al., (2015). Furthermore, forest expansion indicates relatively warm conditions in both the northwestern and southern Iberian Peninsula (Fig. 4B), and are in good agreement with atmospheric temperatures over NE Spain (Fig. 4E) (Martín-Chivelet et al., 2011) suggesting similar on-land conditions across northern Iberia.

However, our records show distinct conditions for the early MCA (900 - 1100 CE) where warm winters, cooler spring-falls and extreme precipitation characterize the Northern Iberian Peninsula (Fig. 5). Increased fluvial input during the MCA, particularly intense around 1000 CE is also observed at other sites in Northern Iberia; Ria de Vigo (Álvarez et al., 2005) and Ria de Muros (Lebreiro et al., 2006). On the contrary, the late MCA (1200 - 1300 CE), shows relatively cooler and stormy winters and warmer spring-falls (Fig. 5).

At present, precipitation occurs mainly as a result of the moisture carried by westerly winds that become predominant during NAO negative phases (Trigo et al., 2004). However, Yang and Myers (2007) have proposed that persistent positive NAO conditions have also generated strong heat and moist transport from the Atlantic.

A clear relationship between the NAO-derived atmospheric circulation over Europe and the decadal variability of north Atlantic surface temperatures (AMO), with northerly winds over Europe associated to a positive state of the AMO and zonal winds to its negative state has been shown by Yamamoto and Palter (2016). But, the clear imprint of AMO variability on European summer temperatures is not observed during wintertime. An absence attributed to a cancelation of the ocean SST expression by strong cold winds (Yamamoto and Palter, 2016).

Previous work from the Iberian/Atlantic ocean region (Abrantes et al, 2011) suggests coherence between SST at Porto and the instrumentally and tree-ring reconstructed AMO implying a connection between the Iberian Peninsula coastal circulation and multidecadal variability of North Atlantic Ocean SSTs (Gray et al., 2004; Mann et al., 2010).

Frankcombe et al. (2010), propose that the AMO is dominated by two main time scales: 20-30 yr associated with the AMOC and so, of ocean internal origin, and 50-70 yr related to the atmospheric exchange between the Atlantic and the Arctic Ocean. More recently, Buckley and Marshall (2016), revised the periodicities shown by various instrumental and proxy records, and grouped them into decadal (20 yr) and multidecadal (± 40-70 yr), and show statistical significance for the 70 yr

periodicity, supporting that ocean dynamics play a significant role in the variability of the European climate on multi-decadal time scales (Yamamoto and Palter, 2016; Zhang, 2007).

To investigate the influence of the (AMO) on the weather regime over the Atlantic Iberian Peninsula, we performed wavelet analysis on our SST and [n-alk] records (Figs. 6, 7). Results reveal a 74 yr periodicity for SST and precipitation on the west coast mainly before 1300 CE and in the south coast after 1580 CE. In addition, longer period processes (100 – 180 yr) are found throughout the 3 SST records and before 1200 CE in the precipitation stack (256 yr). If our wavelet results are interpreted in the light of the above-presented information, multi-decadal variability of the North Atlantic dynamics associated with the Atlantic and Arctic Ocean exchange, has an important impact on SST and precipitation over the Atlantic sector of the Iberian Peninsula, mainly between 900 and 1200 CE in the north and after 1580 CE in the Algarve.

However, to explain the occurrence of big storms clustered in the early MCA on the Northwestern Iberian Peninsula, the modern path of westerly winds under NAO positive conditions would have to be positioned southward of 41ºN, a shift that could have been caused by an increase in mid-latitude blocking anticyclones.

Considering that: (1) SCAND is related to major blocking anticyclones over Scandinavia and has a positive mode associated with above-average precipitation across southern Europe (Comas-Bru and McDermott, 2014; Jerez and Trigo, 2013); (2) the negative relation observed between NAO and precipitation in winter, and the EA and SCAND positive relation to winter precipitation in the Northern Iberian Peninsula; and (3) the slight positive relation of SCAND with precipitation in the north but important negative effect in temperature in the southern Iberian Peninsula in summer (Hernández et al., 2015). One possible explanation for the observed strong precipitation in the north and lower SST in the Algarve during the early MCA may be the effect of a positive SCAND (Fig. 5).

In summary, the interplay between North Atlantic multi-decadal variability associated with the Atlantic and Arctic Ocean exchange and the positive modes of the NAO, EA and SCAND, are inferred to explain extreme precipitation associated with warm winters and cool summers observed in the Iberian Peninsula during the early MCA. On the contrary the cold stormy winters and warm summers of the late MCA are attributed to a shift in the SCAND to a negative mode.

### 5.2.2 The warming of the Industrial Era: atmospheric forcing and oceanographic circulation changes

The transition to the Industrial Era starts at 1750−1850 CE with an increase of SSTs to values similar to those detected prior to 1300 CE in all but the Porto record (Figs. 2A, 5) (Abrantes et al., 2011). The same warming is detectable in all the AP pollen records (Fig. 4B); a pattern that coincides with the atmospheric temperature rise detected in NE Spain, Europe and the Northern Hemisphere (Luterbacher et al., 2016; Martín-Chivelet et al., 2011; Moberg et al., 2005) (Fig. 2D, E, F).

In contrast the SST stack anomaly curve shows a smooth signal when compared to the SST data during the same time frame (Fig. 2A, B). Indeed, the Porto and Algarve records show opposite SST patterns that cancel out and result in a leveled SST in the Stack (Figs. 2A, 2B, 5). Superimposed on the 20$^{th}$ century trend are decadal-scale oscillations of unusually large amplitude but an inverse signal is detected for SST and river input in the Algarve (Fig. 2A, 5). Of particular note is the second rise of SST by the mid 20$^{th}$ century (± 1970) (Fig. 2A), which coincides with the highest global temperatures of the last 1,400 yr (Ahmed et al., 2013) and agrees with a second warming phase in the Western Mediterranean (Lionello et al., 2006).

Zampieri et al. (2016) proposed that the rapid warming periods of the Northern Hemisphere, including the last one in the '90s, are mainly modulated by shifts in the AMO from negative (cold) to positive (warm) phases. A close look to the AMO index records of Gray et al. (2004) and Mann et al. (2010) (Fig. 2H), reveals that both warming steps (1850 and 1970 CE) do indeed occur during warming transitions in the AMO index, supporting the notion of a stronger influence of the North Atlantic SST pattern on southern Iberian climate as found for the central-western Mediterranean Sea (Cisneros et al., 2016).

In the Algarve, the highest SSTs are likely to reflect the warm inner-shelf counter-current associated with large-scale northerly winds during the upwelling season (Garel et al., 2016), which in turn are known to have registered a substantial

intensification during the peak summer months (July to September), in the last 50 yr in SW Iberia (Relvas et al., 2009). Furthermore, the occurrence of periods of strong precipitation when SST is low, appear to be related with the change from a negative mode of the EA between 1900 – 1970, to a positive EA after 1970, while NAO and SCAND remain positive (Fig. 5; NOAA historical archive and indices - http://www.cpc.ncep.noaa.gov/data/teledoc/telecontents.shtml; and

http://www.cgd.ucar.edu/cas/jhurrell/)

In summary, the mid 20[th] century rise in SST in the Algarve emerges as a regional response to the unequivocally warming of the global ocean since the 1960's (IPCC, 2013b). The reverse trend registered at the Porto site compared to the southern records, clearly demonstrates the regional differences that are likely to result from global warming via the complexities of the regional ocean dynamics (Seidov et al., 2017).

Considering the high relevance of such environmental changes to ecosystems organization and sustainability, a more in depth discussion of its effect at the regional scale is necessary.

## 6 Conclusions

The combination of SST and terrestrial input/river discharge records from five sites distributed along the Iberian margin (36º to 42 ºN), captures the spatial distribution of temperature as well as continental precipitation for the last 2,000 yr on different
time scales. Furthermore, the new regional stacks for SST and [n-alk] provide a meaningful form to understand the role of global/hemispheric vs regional variability.

Orbital-scale summer insolation imposes long-term cooling from 0 CE to the beginning of the 20[th] century, at all latitudes with maximum amplitude at the southern sites. Century/decadal scale climate changes follow the overall climatic patterns of the extra-tropical Northern Hemisphere and Europe. The RP and the DA climate records are explained by the mechanisms
proposed by Sánchez-Lopez et al., (2015), MCA shows two climate phases: the early MCA (900-1100 CE) with warm winters, cool summers and extreme flooding imply a link between AMO and a high pressure blocking system over Northwestern Europe (positive-like mode of the SCAND) as well as NAO and EA on a positive phase; the late MCA (1200-1300 CE) cold stormy winters and warm summers suggest a shift in SCAND to a negative mode..

 The LIA is marked by the coldest SSTs and frequent but not extreme storms attributed to a dominant negative NAO and EA
modes. The Industrial Era starts by 1800 CE and is marked with a SST rise in consonance with increasing influence from the internal North Atlantic ocean variability on the Atlantic Iberian Peninsula climate. A second increase in SST at ± 1970 CE is particularly marked at the Algarve site as a regional imprint of the global warming impact previously simulated for southern Iberia.

**7. Team list**
**8. Copyright statement**
**9.** Code availability
10..**Data availability**
    Primary data for the new sedimentary sequences is archived in PANGEA (doi: 10.1594/PANGAEA.882269).

**11. Appendices**
**12. Author contribution**

Abrantes, F – PI of the various projects that funded all the data combined in this paper, had the idea and wrote the paper;

Rodrigues, T– Responsible for the biomarkers analysis in all cores;

Rufino, M – Statistical data analysis;

Naughton, F – Responsible for the pollen interpretation and age-models of DIVA and POPEI;

Salgueiro, E – Data of GeoB11033-1 core;

Oliveira, D – Pollen analysis of the DIVA core;

Domingues, S – Pollen analysis for the Tejo site D13882 and POPEI core;

Costa, A. – Age-model for core GeoB11033-1;

Oliveira, P – Processed the satellite-derived and OISST data;

Drago, T – Provided the Algarve core (POPEI);

Mil-Homens, M – Participated in the DIVA cruise;

**Acknowledgments**

The authors express their gratitude to the captain, crew and participants of the cruises Discovery 249, Poseidon PALEO I, GALIOMAR P34, and B/O Mytilus DIVA09 for their contribution during the retrieval of the various cores used in this study. Thanks are due in particular to Guilhermo F. Pedraz and the Departamento de Geociencias Marinas y Ordenación del
Territorio (Universidade de Vigo) for allowing the recovery of the DIVA core on their DIVA09 cruise. Special acknowledgments are due to the anonymous referees of previous versions of this paper, whom have greatly contributed for its improvement.

Funding was provided by projects INGMAR (FCT ARIPIPI Program –Support for State Labs Development), HOLSMEER (EVK2-CT-2000-00060), CLIMHOL (PTDC/AAC-CLI/100157/2008), MIÑO-MINHO (0234_NATURA_MM_1_E),
POPEI (PDCT/MAR/55618/2004), CALIBERIA (PTDC/MAR/102045/2008 from FCT and COMPETE/FEDER -FCOMP-01-0124-FEDER-010599), CIIMAR (20132017 CIMAR), CCMAR (Associated Lab PEstC/ MAR/LA0015/2013); felowships to Filipa Naughton (SFRH / BPD / 36615 / 2007), Teresa Rodrigues (SFRH/BPD/66025/2009), Emília Salgueiro (Ref. SFRH / BPD / 26525 / 2006 & SFRH / BPD /111433 / 2015). Marta Rufino was funded by *contrato ciência* 2007 and by a post- doctoral grant of IPMA, within the EU project SAFI (FP7- SPACE-2013-1, grant agreement n8 607155). Finally
we thank A. Inês, D. Ferreira and C. Monteiro for their help and support with the laboratory analysis, and Zuzanna Stroynowski by the English revision.

**List of Tables and Figures**

Table 1 – ID, water depth, geographic location, core type, sampling cruise, sedimentation rate (SR), and age-model origin of the seven sedimentary sequences used in this study.

Table 2 –Mean Uk'$_{37}$ –SST at the Iberian Peninsula margin during major climatic periods of the last 2,000 yr.. Shades of blue and pink highlight respectively colder and warmer periods. * Existing sediment hiatus likely affected estimated value.

Table 3 – Compilation of flooding and drought events on the Western Iberian Peninsula, according to published information
for the Medieval Climate Anomaly (MCA), the Little Ice Age (LIA) and the Modern/ Industrial Era. *Italic* refers to document source..

Figure 1– Cores location over Iberian Margin bathymetry. Geoß11033-1 / Galiza in orange, DIVA09 GC / Minho in magenta, PO287-6B, -6G / Porto (Douro Mud Belt) in blue, PO287-26B, -26G, D130902, D13882 Tejo (Tagus Mud Belt) in
green, and POPEI VC2B/ Algarve in red. The color assign to each core location will be applied in all figures (A). Winter satellite-derived sea surface temperature (SST) image (23 Jan 2003) showing the surface signature of the Iberian Poleward Current (B); Summer SST image illustrating coastal upwelling conditions (colder coastal SST) along the Western Iberian Margin (3 Aug 2005) (C). B and C land covered areas appear in white, the color shades were selected to highlight the main oceanographic structures, the image source is: JPL-PODAAC https://mur.jpl.nasa.gov/

Figure 2 – Comparison of Uk'$_{37}$-SST along the last 2,000 years at sites Galiza (orange), Minho (magenta), Porto (blue), Tejo (green) and Algarve (red) (A); SST stack constructed from all Iberian margin records (B); Uk'$_{37}$-SST at Ria de Vigo (Diz et al., 2003) (C); Northern Hemisphere annual mean atmospheric temperature anomaly (Moberg et al., 2005) (D); Northern Spain atmospheric temperature anomaly (Martín-Chivelet et al., 2011) (E); European spring-fall atmospheric temperature anomaly (Luterbacher et al., 2016) (F); volcanic activity as aerosol optical depth (AOD) (Crowley and Unterman, 2013) (G); radionuclide-derived total solar irradiance (TSI) (Bard et al., 2007) (H); Northern Atlantic Ocean SST anomaly, AMO index (gray, (Mann et al., 2010) and (black, Gray et al., 2004) (I); NAO index (Luterbacher et al., 2002); brown - (Trouet et al., 2009) (J). Light grey band marks the Roman Period (RP), the pink band marks the Medieval Climate Anomaly (MCA); the blue band marks the Little Ice Age (LIA). DA – Dark ages.

Figure 3 – [n-alk] (ng/g) variability along the last two millennia and the 5 Iberian margin sites (A), colors as in Figure 3; the [n-alk] stack anomaly (B); total pollen concentration (TPC, nº pollen grains/cm$^3$ sediment) (C); SST stack anomaly (D); SST NW Atlantic cores KNR140-2-59GGC and CH07-98-MC22 (Saenger et al., 2011) (E); NAO index (black line, Luterbacher et al. 2002), (green line; (Cook et al., 2002), (gray line; Trouet et al, 2009), (brown line; Ortega et al., 2015) (F). Dark grey bands mark the periods of Atlantic flooding as listed in Table 3. Light grey band marks the Roman Period (RP), the pink band marks the Medieval Climate Anomaly (MCA); the blue band marks the Little Ice Age (LIA). DA – Dark ages.

Figure 4 – Variability of semi-desert plants percentages along the 2,000 yr in cores Diva, Tejo, Popei (magenta, green and red respectively) and Ria de Vigo (black) (Desprat et al., 2003) (A); arboreal pollen percent abundance (B); the SST stack (C); the [n-alk] stack (D); Northern Spain atmospheric temperature anomaly (Martín-Chivelet et al., 2011) (E); European spring-fall atmospheric temperature anomaly (Luterbacher et al., 2016) (F); Spring Precipitation Central Europe (Büntgen et al., 2011) (G): European Seasonality (H) and Spring AT (I) (Luterbacher et al., 2004); May-August Precipitation (Touchan et al., 2005) (I); winter (DJFM) Precipitation (Romero-Viana et al., 2011) (K). Light grey band marks the Roman Period (RP), the pink band marks the Medieval Climate Anomaly (MCA); the blue band marks the Little Ice Age (LIA). DA – Dark ages.

Figure 5 – Sea Surface Temperature (SST) and river-input ([n-alk] STACKS for NW Iberia (Galiza, Minho e Porto) and continental T an humidity conditions extracted from pollen Arboreal Pollen (% AP) at Minho, are compared to the Algarve SST and [n-alk] anomalies (for easier comparison) and % AP. Pink bars mark the warmer periods (Roman Period- RP; Medieval Climate Anomaly – MCA; Industrial Era- IE), and the blue bar marks the cold Little Ice Age (LIA). Periods of increased AMO impact on either IP region are marked by AMO. The dominant state of the Atmospheric Circulation Modes (ACM) in the North Atlantic for the major climatic periods of the last 2,000 yr is depicted on the top of the figure (North Atlantic Oscillation- NAO; East Atlantic- EA; Scandinavia- SCAND). The two consecutive signs marked for IE correspond to the early IE, 1850-1970, and a late IE, after 1970 CE.

Figure 6 – The continuous wavelet power spectrum of the SST STACK (A); the north SST STACK (B); and the Algarve SST record (C). The thick black contour designates the 95% confidence level and the lighter shaded area represents the cone of influence (COI) where edge effects might distort the results.

Figure 7 – The continuous wavelet power spectrum of the [n-alk] Stack record (A); the north [n-alk] Stack (B) and the Algarve [n-alk] record (C). The thick black contour designates the 95% confidence level and the lighter shaded area represents the cone of influence (COI) where edge effects might distort the results.

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

| Site | ID | Water depth (m) | Lat N | Long W | Core Type | Cruise | SR (mm/yr) | Age Model |
|------|-----|------|--------|---------|-----------|--------|-----------|-----------|
| Galiza | GeoB11033-1 | 1873 | 42.1698 | -9.5360 | Box | Galiomar P342 | 0.4 | This work |
| Minho | DIVA09GC | 119 | 41.9168 | -9.0735 | Gravity | Sarmento de Gamboa | 0.5 | This work |
| Porto | PO287-6B, 6G | 84 | 41.3356 | -8.9888 | Gravity | RV Poseidon - PALEO1 | 6.3 | Abrantes et al., (2011) |
| Tejo | D13902 | 90 | 38.5540 | -9.3355 | Long Piston | RV Discovery 249 | 7.0 | Abrantes, F.,et al, (2005) |
| Tejo | PO287-26B, 26G | 96 | 38.5582 | -9.3640 | Box/ Garvity | RV Poseidon - PALEO1 | 7.0 | Abrantes, F.,et al, (2005) |
| Tejo | D13882 | 88 | 38.6450 | -9.4542 | Long Piston | RV Discovery 249 | 0.2 | Rodrigues et al, (2009) |
| Algarve | POPEI VC2B | 96 | 36.8800 | -8.0700 | Vibrocore | NRP Auriga - POPEI0108 | 1.2 | This work |

**Table 1 –ID, water depth, geographic location, core type, sampling cruise, sedimentation rate (SR), and age model origin of the seven sedimentary sequences used in this study.**

| | Common Era | Mean SST | | | | |
|---|---|---|---|---|---|---|
| | | Galiza | Minho | Douro | Tejo | Algarve |
| Roman - Dark Ages | < 900 | 16.0 | 16.5 | - | 17.5 | 19.1 |
| Medieval Warm Period | 900 - 1300 | 16.6 | 16.2 | 15.3 | 17.1 | 18.7 |
| Little Ice Age | 1350 -1850 | 16.1 | 15.9 | 14.7 | 15.6* | 17.8 |
| Modern Times | >1900 | 15.5 | - | 14.7 | 15.5 | 18.3 |

**Table 2 –Mean Uk'$_{37}$-SST at the Iberian margin during the major climatic periods of the last 2,000 yr. Shades of blue and pink highlight respectively colder and warmer periods.**
**\*Existing sediment hiatus likely affected estimated value.**

**FLOODS**

| Period/ Region | Atlantic Basin | Douro & Minho | Tejo | Guadiana | References | Oservations |
|---|---|---|---|---|---|---|
| MCA | | | **785-1205** | | Benito et al, 2003 | Increase magnitude and frequency |
| | | *1000 - 1100* | | | *Tullot, 1988* | *Doc Sources* |
| | **1000-1200** | | | | Benito et al, 2003 | Sediment records |
| | | | *1150 -1200* | | *Benito et al, 2010* | *Max from Doc sources* |
| | 1150 -1290 | | | | Benito et al, 2003 | Sediment records |
| | | *1180 – 1200* | | | *Tullot, 1989* | *Doc Sources* |
| LIA | **1430-1685** | | | | Benito et al, 2010 | Sediment records & Doc Sources |
| | | *1434 - LCF* | | *1434 - LCE* | *Barriendos and Rodrigo, 2010* | *Doc Sources* |
| | | *1450-1470* | | *1450-1500* | *Tullot, 1988* | *Doc Sources* |
| | | | 1450 -1500 | | Benito et al, 2003 | High frequency lower magnitude |
| | | *1545 - LCF* | | *1545 - LCE* | *Barriendos and Rodrigo, 2010* | *Doc Sources* |
| | | | | *1570-1630* | *Barriendos and Martín-Vide, 1998* | *Mediterranean area- Doc Sources* |
| | *1590-1610* | | | | *Benito et al, 2004* | *Doc Sources* |
| | | ***1626- VLCF*** | ***1626- VLEF*** | ***1626- VLCF*** | *Barriendos and Rodrigo, 2010* | *Doc Sources* |
| | | *1636 - LCF* | *1637 - LCE* | | *Barriendos and Rodrigo, 2010* | *Doc Sources* |
| | ***1730-1810*** | | | | *Benito et al, 2010* | *Doc Sources* |
| | 1730-1760 | | | | Benito et al, 2003, 2004 | Sediment records & Doc Sources |
| | | | | *1830-1870* | *Barriendos and Martín-Vide, 1998* | *Mediterranean area- Doc Sources* |
| | | *1778 - LEF* | | *1778 - LCE* | *Barriendos and Rodrigo, 2010* | *Doc Sources* |
| | *1780-1810* | | | | *Benito et al, 2004* | *Doc Sources* |
| MODERN | | *1853 - LEF* | | | *Barriendos and Rodrigo, 2010* | *Doc Sources* |
| | | *1860 - LCF* | *1860 - LCF* | | *Barriendos and Rodrigo, 2010* | *Doc Sources* |
| | 1870-1900 | | | | Benito et al,  2003 | Sediment records |
| | 1930-1950 | | | | Benito et al, 2003 | Sediment records |
| | 1960-1980 | | | | Benito et al, 2003 | Sediment records |
| | | | **1670 - 1950** | 1876 | Benito et al, 2003 | High frequency lower magnitude |
| | | | *1950-1980* | *1979* | *Benito et al, 2004* | *Doc Sources* |

LCF - Large Catastrophic Flood; LEF - Large Extraordinary Flood; VLCF - Very Large Catastrophic Flood; VLEF - Very Large Extraordinary Event

Bold - Catastrophic Event

**DROUGHTS**

| | | | | | | |
|---|---|---|---|---|---|---|
| MWP | | | | 1361-1390 | Barriendos and Martín-Vide, 1998 | |
| LIA | | | | 1511-1540 | Barriendos and Martín-Vide, 1998 | |
| | 1540-1570 | | | | Barriendos 2002 | severe |
| | 1625-1640 | | | | Barriendos 2002 | severe |
| | 1750-1760 | | | | Barriendos 2002 | Less severe |
| | 1810-1830 | | | | Barriendos 2002 | Less severe |
| | | | | *1880-1950* | *Barriendos and Martín-Vide, 1998* | |
| MODERN | 1880-1910 | | | | Barriendos 2002 | Less severe |

**Table 3 – Compilation of flooding and drought events on the Western Iberian Peninsula, according to published information for the Medieval Climate Anomaly (MCA), the Little Ice Age (LIA) and the Modern/ Industrial Era. *Italic* refers to document source.**

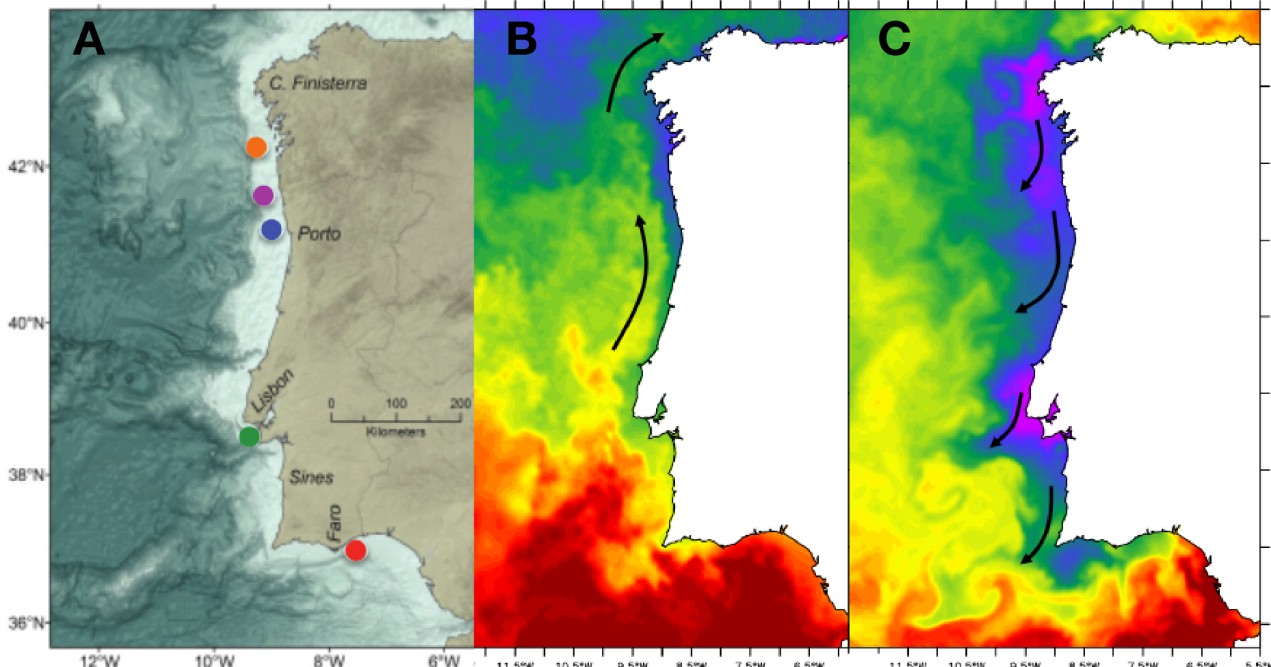

**Figure 1– Cores location over Iberian Margin bathymetry. Geoß11033-1 / Galiza in orange, DIVA09 GC / Minho in magenta, PO287-6B, -6G / Porto (Douro Mud Belt) in blue, PO287-26B, -26G, D130902, D13882 Tejo (Tagus Mud Belt) in green, and POPEI VC2B/ Algarve in red. The color assign to each core location will be applied in all figures (A). Winter satellite-derived sea surface temperature (SST) image (23 Jan 2003) showing the surface signature of the Iberian Poleward Current (B); Summer SST image illustrating coastal upwelling conditions (colder coastal SST) along the Western Iberian Margin (3 Aug 2005) (C). B and C land covered areas appear in white, the color shades were selected to highlight the main oceanographic structures, the image source is: JPL-PODAAC https://mur.jpl.nasa.gov/**

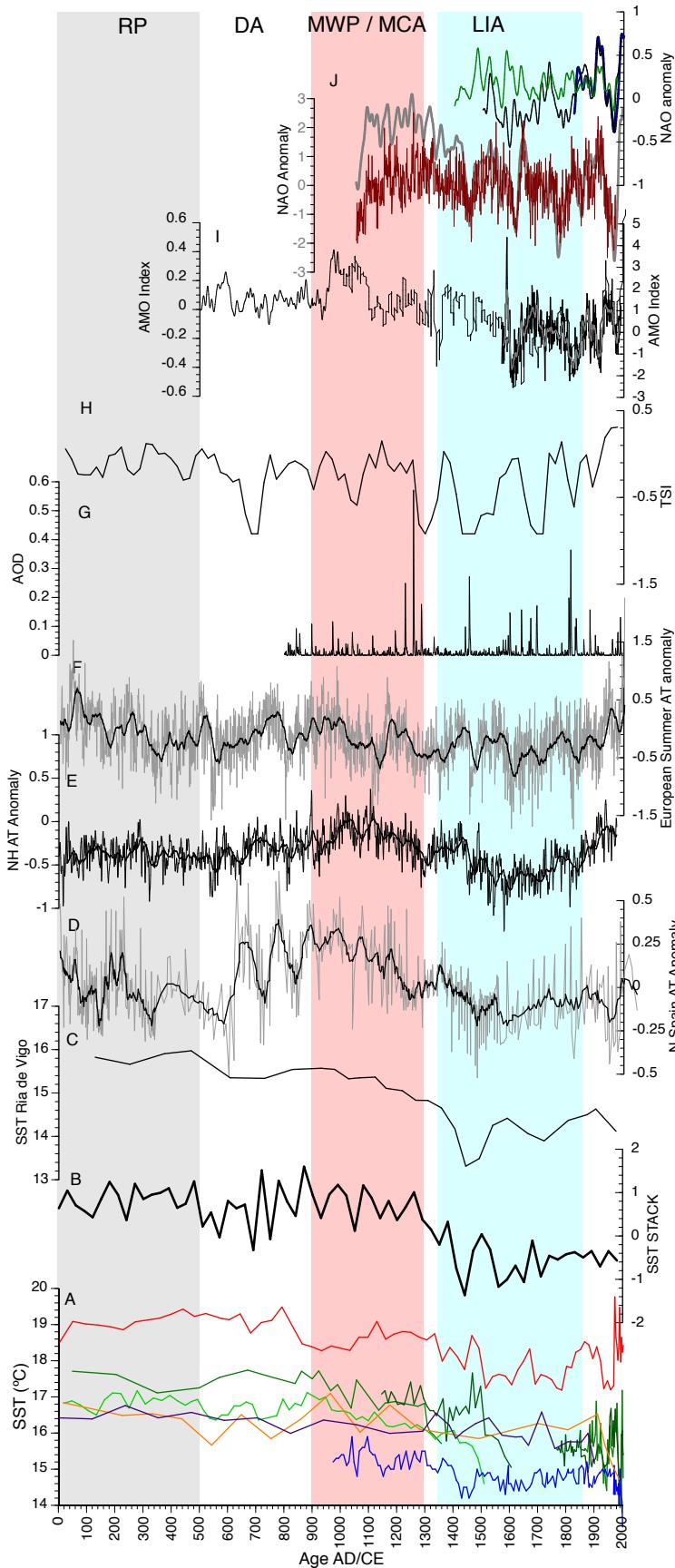

**Figure 2 – Comparison of Uk'₃₇-SST along the last 2,000 years at sites Galiza (orange), Minho (magenta), Porto (blue), Tejo (green) and Algarve (red) (A); SST stack constructed from all Iberian margin records (B); Uk'₃₇-SST at Ria de Vigo (Diz et al., 2003) (C); Northern Hemisphere annual mean atmospheric temperature anomaly (Moberg et al., 2005) (D); Northern Spain atmospheric temperature anomaly (Martín-Chivelet et al., 2011) (E); European spring-fall atmospheric temperature anomaly (Luterbacher et al., 2016) (F); volcanic activity as aerosol optical depth (AOD) (Crowley and Unterman, 2013) (G); radionuclide-derived total solar irradiance (TSI) (Bard et al., 2007) (H); Northern Atlantic Ocean SST anomaly, AMO index (gray, (Mann et al., 2010) and (black, Gray et al., 2004) (I); NAO index (Luterbacher et al., 2002); brown - (Trouet et al., 2009) (J). Light grey band marks the Roman Period (RP), the pink band marks the Medieval Warm Period/ Medieval Climate Anomaly (MWP/MCA); the blue band marks the Little Ice Age (LIA). DA – Dark ages.**

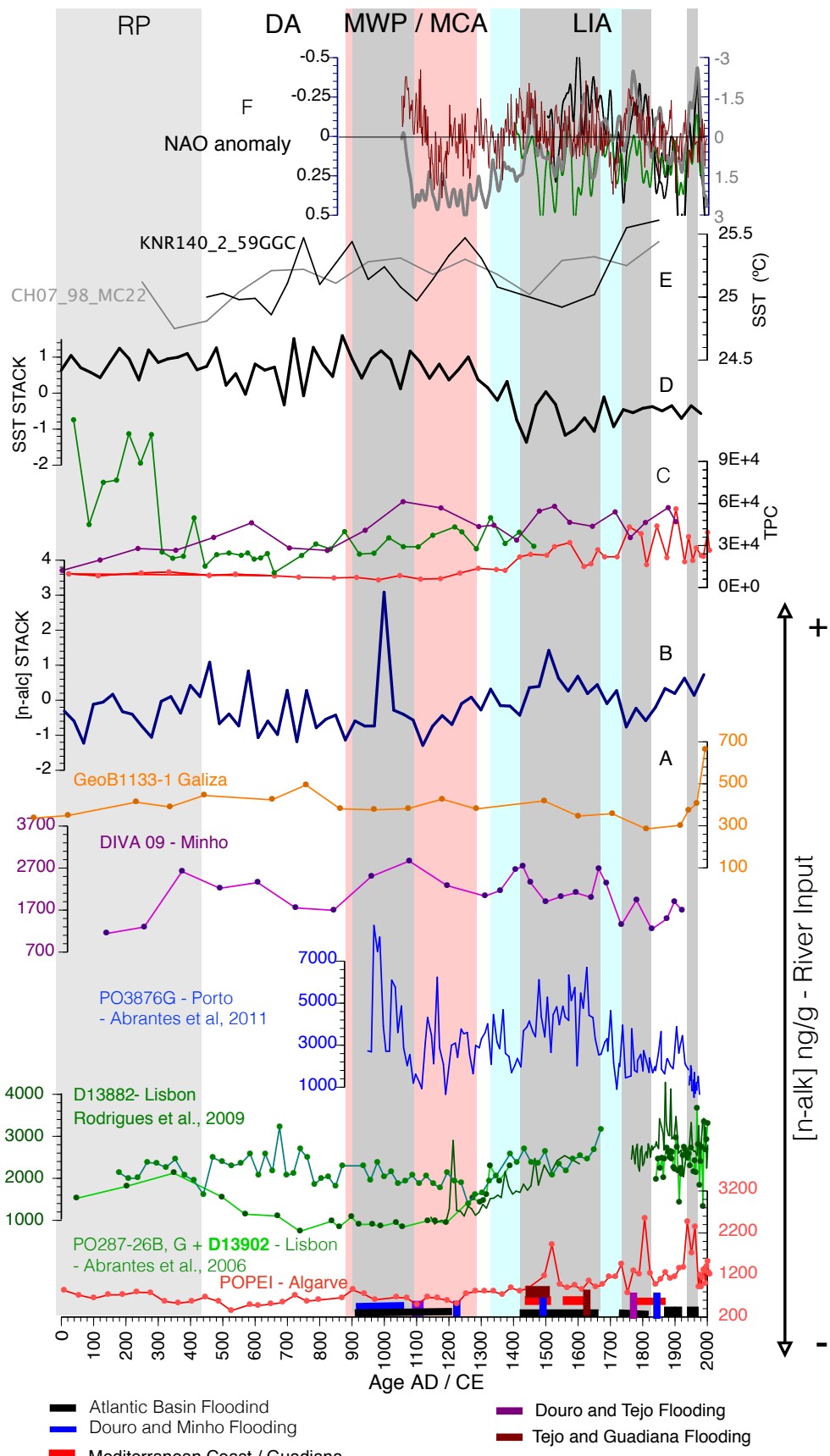

**Figure 3 – [n-alk] (ng/g) variability along the last two millennia and the 5 Iberian margin sites (A), colors as in Figure 3; the [n-alk] stack anomaly (B); total pollen concentration (TPC, nº pollen grains/cm³ sediment) (C); SST stack anomaly (D); SST NW Atlantic cores KNR140-2-59GGC and CH07-98-MC22 (Saenger et al., 2011) (E); NAO index (black line, Luterbacher et al. 2002), (green line; (Cook et al., 2002), (gray line; Trouet et al, 2009), (brown line; Ortega et al., 2015) (F). Dark grey bands mark the periods of Atlantic flooding as listed in Table 3. Light grey band marks the Roman Period (RP), the pink band marks the Medieval Warm Period/ Medieval Climate Anomaly (MWP/MCA); the blue band marks the Little Ice Age (LIA). DA – Dark ages.**

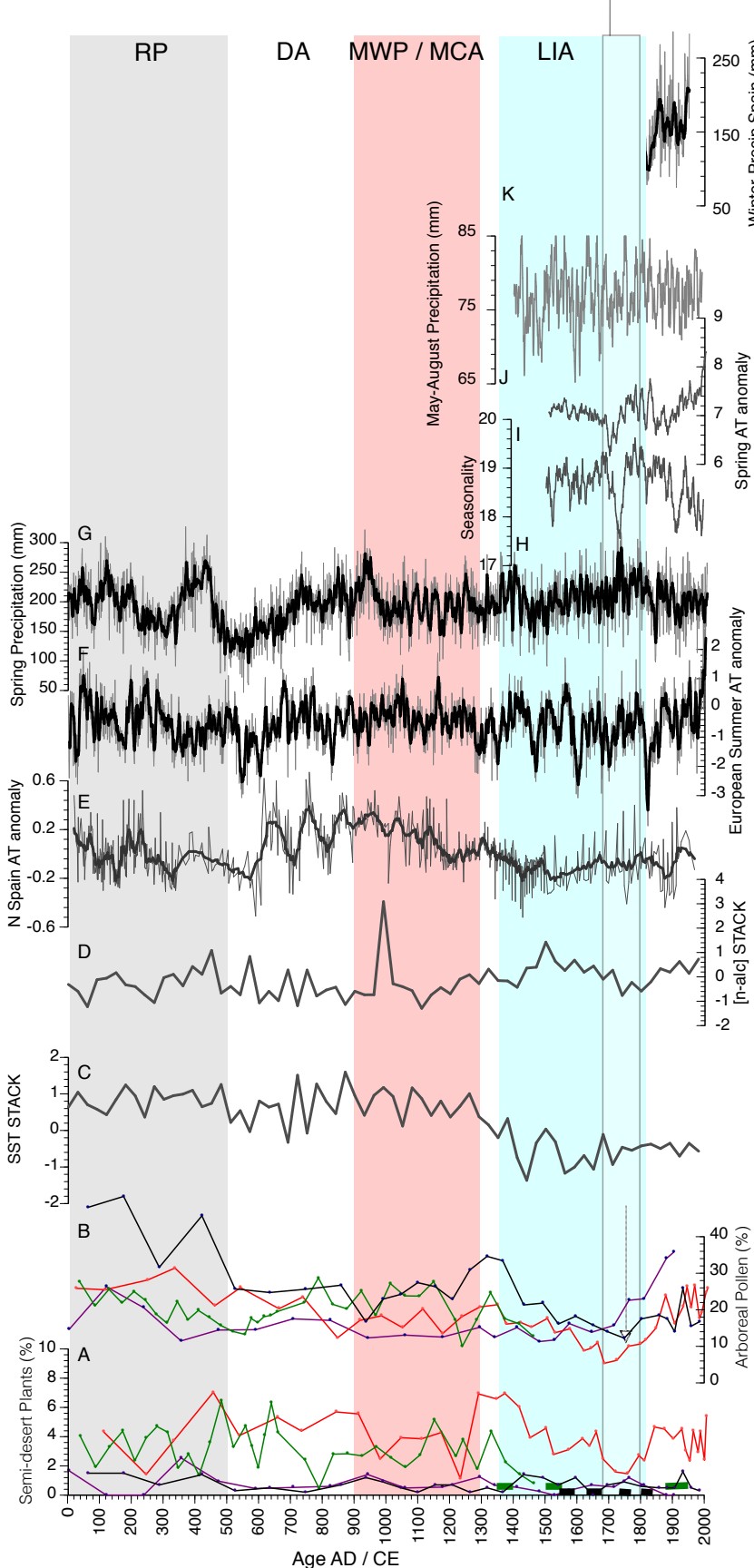

**Figure 4 – Variability of semi-desert plants percentages along the 2,000 yr at Minho, Tejo, Algarve (magenta, green and red respectively) and Ria de Vigo (black) (Desprat et al., 2003) (A); arboreal pollen percent abundance (B); the SST stack (C); the [n-alk] stack (D); Northern Spain atmospheric temperature anomaly (Martín-Chivelet et al., 2011) (E); European spring-fall atmospheric temperature anomaly (Luterbarher et al., 2016) (F); Spring Precipitation Central Europe (Büntgen et al., 2011) (G): European Seasonality (H) and Spring AT (I) (Luterbacher et al., 2004); May-August Precipitation (Touchan et al., 2005) (I); winter (DJFM) Precipitation (Romero-Viana et al., 2011) (K). the pink band marks the Medieval Warm Period/ Medieval Climate Anomaly (MWP/MCA); the blue band marks the Little Ice Age (LIA). DA – Dark ages.**

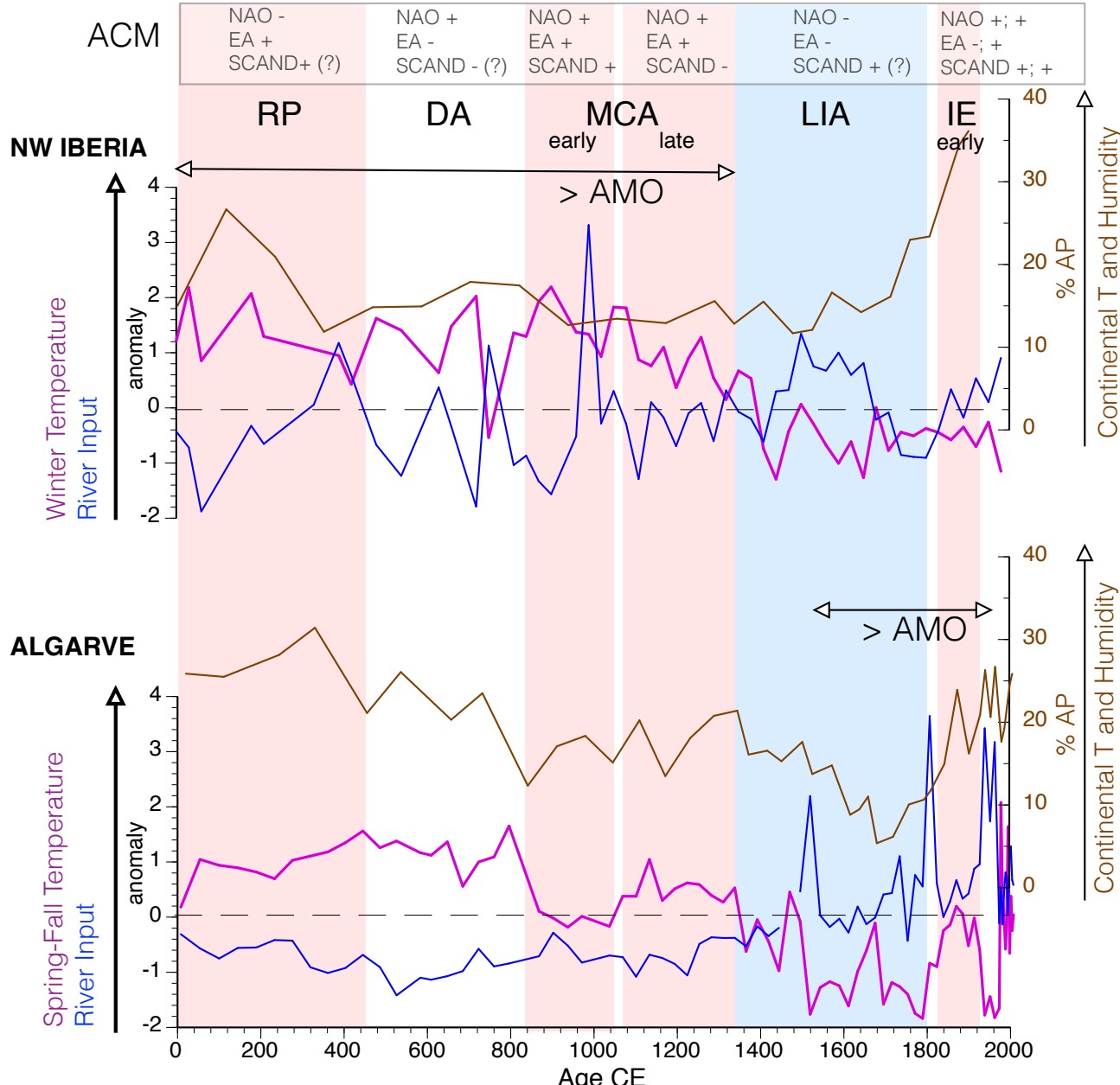

**Figure 5 – Sea Surface Temperature (SST) and river-input ([n-alk] STACKS for NW Iberia (Galiza, Minho e Porto) and continental Temperature (T) and humidity conditions extracted from pollen Arboreal Pollen (% AP) at Minho, are compared to the Algarve SST and [n-alk] anomalies (for easier comparison) and % AP. Pink bars mark the warmer periods (Roman Period-RP; Medieval Climate Anomaly – MCA; Industrial Era- IE), and the blue bar marks the cold Little Ice Age (LIA). Periods of increased impact of the Atlantic Multidecadal Oscillation on either NW and Algarve regions of the Iberian Peninsula are marked by AMO. The modes of the dominant North Atlantic Atmospheric Circulation Modes (ACM) during the major climatic periods of the last 2,000 yr are depicted on the top of the figure (North Atlantic Oscillation- NAO; East Atlantic- EA; Scandinavia- SCAND). The two consecutive signs marked for IE refer to the early IE (1850-1970), and a late IE (after 1970 CE).**

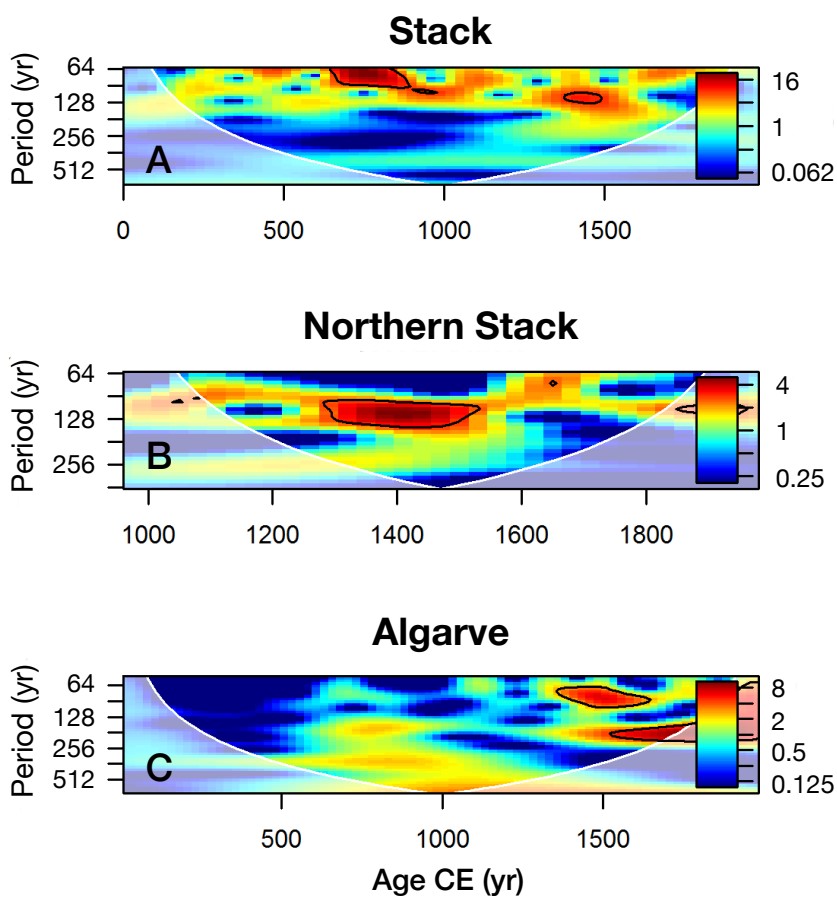

**Figure 6** – The continuous wavelet power spectrum of the SST STACK (A); the north SST STACK (B); and the Algarve SST record (C). The thick black contour designates the 95% confidence level and the lighter shaded area represents the cone of influence (COI) where edge effects might distort the results.

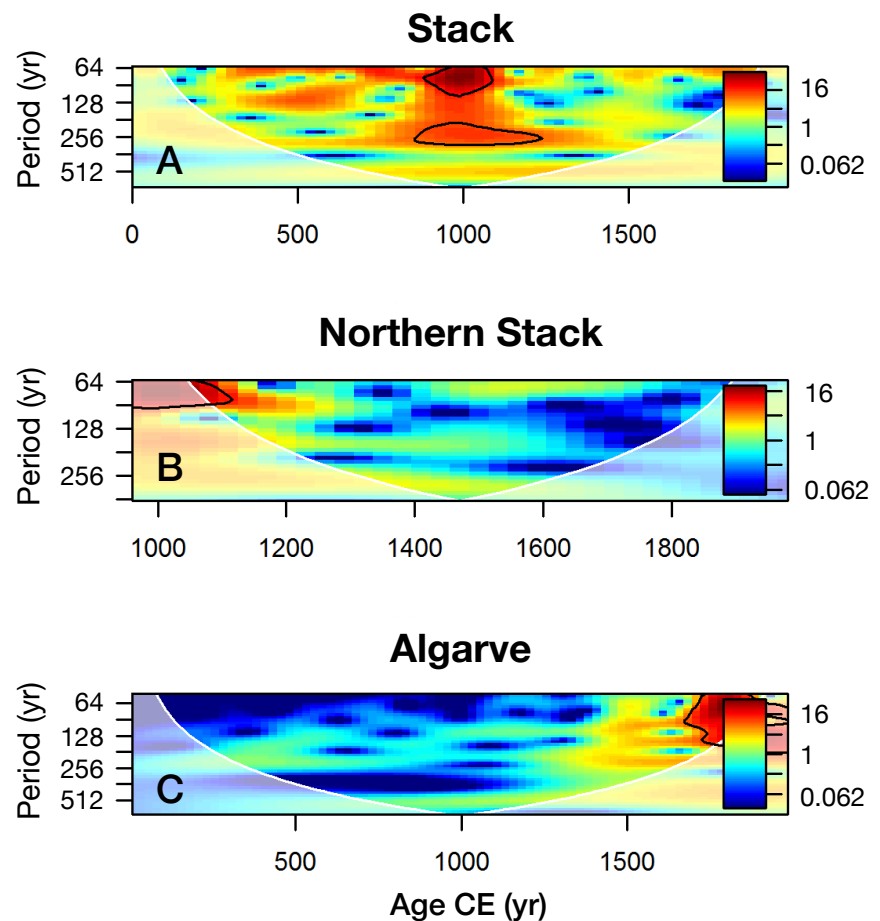

**Figure 7 –** The continuous wavelet power spectrum of the [n-alk] STACK record (A); the north [n-alk] STACK (B) and the Algarve [n-alk] record (C). The thick black contour designates the 95% confidence level and the lighter shaded area represents the cone of influence (COI) where edge effects might distort the results.