# Peer review of "The Climate of the Common Era off the Iberian Peninsula"

_Climate of the Past, 2017_

## Referee Comment (RC1) · Anonymous Referee #1 · 12 Sep 2017

The paper of Abrantes et al.: "The Climate of the Common Era off the Iberian Peninsula" is discussing five sites distributed along the Iberian margin spanning the last 2000 years. A multi-proxy approach was chosen here where alkenone derived Sea Surface Temperature (SST) reconstructions were compared to on-land precipitation given by higher plant n-alkanes and pollen data.

In general, the records provide useful information about climate variability offshore the Iberian Peninsula over the common area. However, I feel the paper is too long and partly confusing as the message is not clear. It is not evident to me, how in particular the changes in the ocean temperature observed would drive or not directly precipitation changes on land? I would suggest that the author should shorten the paper to the main findings and concentrate on the question: what is the main message of this paper to

the community?

The paper should definitely be proof read by a native English speaker, as many parts of the paper are very hard to understand lacking a sentence structure and words.

I struggle with figure 4, 5 and 6 as these contain way too many plots and I find it hard to see the link between different data sets and their correlation as described in the text.

Age model: The 3 new age models of the cores should be shown as an age-depth plot additionally to the table with the 14C dates. Moreover, a Bayesian age depth model should be performed to better constrain age uncertainties.

Some specific comments below: Page 1 line 18: The Iberian Peninsula, at North Atlantic mid-latitude and the western extreme of the European continent, is a relevant area for climate reconstructions. – Rephrase sentence and what makes it a relevant area for climate reconstructions? Line 25: Is that even significant as the calibration error on alkenone SST is 1.5 C? Schouten et al., 2013, http://dx.doi.org/10.1016/j.orggeochem.2012.09.006

Page 2 Line 2: change to Medieval climate anomaly Line 5: what does particular mean? Line 7: "The intense precipitation/ flooding and warm winters but cooler intermediate seasons (spring and fall) observed for the early MWP imply the interplay of internal oceanic variability with the three atmospheric circulation modes, North Atlantic Oscillation (NAO), East Atlantic (EA) and Sandinavia (SCAND) in a positive phase".- how would the interplay of these 3 patterns cause the observed pattern? Line 15: rephrase-sentence like that makes no sense Line 32: restructure Line 33: delete Medieval Warm Period (MWP)

Page 3: Line 27: rephrase bad English

Page 4 Line 23-26: superficial statement needs more explanation Line 30: change to: For that we combine the above mentioned published records with 3 new records located along the Iberian margin from 42° N to 36 °N, covering the last 2,000 yr

Page 6: Line 4: Any additional proof that the cores are tracing river input despite pollen like BIT index

Page 12 Line 17-18: not clear Line 32: what does that mean important decrease?

Page 13 Line 15: what does the N stands for? Line 19: rephrase

Page 17 Line 20: Specific climate conditions – unclear what does specific indicate?

---

## Referee Comment (RC2) · Anonymous Referee #2 · 27 Sep 2017

The article by Abrantes et al. both reviews the available literature on climate changes around the Iberian Peninsula during the common era and analyze a series of multi-proxy/multi-site records of SST and continental runoff.

I feel the article is too long and descriptive, and clearly lacks a problematic. I wonder if it would be more suitable either to shorten it thoroughly - to only present the new results - or to write a kind of two-part article (part I presenting the review /Âǎstate of the art, part II presenting the new results). On the one hand, the authors are clearly the most appropriate paleoclimatologists to provide an extensive review of the Iberian Margin's paleoclimatology during the common era, and it would be a shame if their knowledge could not be shared with the community, but on the other hand I find that the article is really difficult to follow as it stands. Anyway, the authors should distinguish more clearly

those parts of the article where the new data are presented and analyzed WRT those other parts where the new data are confronted to the literature. The 'review' aspect of the article is disseminated everywhere in the article, and too often shows up without further justification, which sometimes leads to non-sequitur.

It is much easier for me to review the article by listing specific comments:

-page 1: Abstract, first sentence, is not convincing at all. Please remove it.

-page 2: Lines 6-11, those lines are too complex and could not be properly understood without having a read over the modern climatology chapter.

-page 2: Line 23, perhaps cite Guiot and Cramer, 2016, Science, for a more recent assessment.

-page 3: Here the discussion would greatly benefit if the authors could add a series of very simple figures introducing the NAO, EA and SCAND modes of climate variability, in particular since the authors often refer to those modes later in the discussion.

-page 4: Here the reader is really lost, and could not remember any clear information at the end of the page.

-page 6: Line 31, please check that ''standardized" and ''scaled" are not referred to ''normalized" and ''standardized" instead.

-page 8: Line 2, ''All age models … all accepted 14C dated levels" reads like you've discarded some of them. Please clarify the age model description.

-page 9: Lines 23-31, the discussion on the most recent SST shifts could be either discussed later, or more developed (what is the great salinity minimum?). It is difficult to see what happens over the last 50 years.

-page 10: on the n-alcane concentrations, lines 1-10 please explain more how you calibrate the proxy. I would intuitively expect that dilution plays an important role, so that the more riverine runoff you get, the more alcanes would be diluted by terrigenous

material, but it seems to be the contrary. . .

-chapter 5.3: please try to be more concise through sorting out the results and discussion separately. Why are you not discussing the LIA? Also, I find the wavelet analysis neither convincing nor useful to the discussion, and I don't see how you could extract significant periodicities longer than a century over time windows shorter than two centuries.

-figures: please check the captions. In general, there are too many panels.

Overall, the article remains too confusing at this stage. I realize it is difficult to be constructive, and that I share many concerns with the Anonymous Referee #1. Perhaps the key would be to frame better the manuscript, including the identification of a clear scientific question, along with clearer chapters and sub-chapters within which diverse informations do not impinge from one chapter to another.

---

## Author Comment (AC1) · 12 Nov 2017

Please consider reading the supplement for a more clear vision of the reply and the new abstract.

The authors thank referee 1 for his important contribution to improve the manuscript. In general, the reviewer considers that the presented records provide useful information about climate variability offshore the Iberian Peninsula over the Common Era. However, the reviewer finds the paper too long and unable to pass a clear message, and suggests the paper to concentrate on answering a clear question. The length of the paper has been substantially reduced. The introduction was shortened, the material and methods was reduced to the essential information and most of the detailed and

considered important information is now compiled as Supplementary material. The essential information relative to the cores chronology was included in the methods and the individualized chronology section of the previous version was deleted. Detailed information on the age-model construction for the new sedimentary sequences is now also included in the supplementary material. The results and discussion section was subdivided and results are now presented separately. The discussion has also been re-organized around the specific questions raised by the data. Abstract and Conclusions have equally been re-written in what we hope to be a more concise style. We certainly hope that the re-organization of the paper makes it easier to read and helps to better convey the message(s) included.

The reviewer considers also that the paper should definitely be proof read by a native English speaker, as many parts of the paper are very hard to understand lacking a sentence structure and words. The new and much changed version has been thoroughly revised by a native English speaker. Age model: The 3 new age models of the cores should be shown as an age-depth plot additionally to the table with the 14C dates. Moreover, a Bayesian age depth model should be performed to better constrain age uncertainties. An explanation and data used for the definition of the age-models for the new three cores is now included in the supplementary material. However, in order to correctly respond to this comment, below we present a discussion on the methodology used for the age model construction of all 7 sedimentary sequences used in this paper, the comparison between methods and the basic data for the three new sedimentary sequences, including de age-depth models.

Chronology: The example of PO287-6 The age-model for the spliced sequence composed of cores PO287-6B and 6G (box, gravity) was constructed by combining two methods: (1) 210Pb activity measured in box-core samples (Fig. 1A) which depending on the accepted model provides a sedimentation rate varying between 0.32 and 0.43 cm yr-1; (2) four accelerator mass spectrometry (AMS) radiocarbon measurements (Leibniz-Laboratory for Radiometric Dating and Stable Isotope Research, Kiel,

Germany) (Table 1). Two further ages were assigned through MS correlation to other well-dated cores recovered off Lisbon (Fig. 1B). Raw AMS 14C dates were corrected for reservoir effect by 400 yr (Abrantes et al., 2005) and converted to calendar ages with the INTCAL04 data set (Reimer et al., 2004). The obtained calendar ages are presented in years Anno Domini (AD/CE). To develop a continuous record, the splicing of the long cores (piston and gravity) with the box-core (PO287-6B, 6G) was done through the Magnetic Susceptibility record (MS) of both cores (Fig. 1B). Further integration of the above-referred cores was based on the 1952 CE age found at 20.7 cm (depth corrected for compaction during sub-sampling) in box-core PO287-6B. Comparison of the PO287-6G MS record to sedimentary sequences from the Tagus system (Abrantes et al. 2005) was also done; Figure 1C depicts Depth vs. AD ages (with $2\sigma$ error) for PO287-6G with a linear best fit. An age that is within the error of the age estimated for the same depth using the sedimentation rate that results from a linear interpolation of the five considered levels (Table 1, Figs. 1B, 1C). Given the uncertainty associated to the 14C dates, the establishment of an age model based on the interpolation between each dated level is normally avoided for sequences covering short time intervals (Jan Heinemeier, pers com.). An age/depth relationship defined by the linear best-fit line of the calibrated 14C ages is the most common approach (e.g. (Narayan et al., 2010)). However we decided to compare age-depth models using both a linear and a polynomial best fit for core PO287-6G (Fig. 2). Both models give very close ages on the interval with dated levels, but the lack of dates at the base of core PO287-6G leads to older ages at the bottom of the record when using the polynomial solution. Why the selection of a linear interpolation? The assumption of a constant sedimentation rate was applied in Abrantes et al., 2005 (QSR) following the advise of Jan Heinemeier (Aharus University 14C dating center). According to this expert, in the case of records covering short time-scales, such as the last 2,000 yr, and with a relatively small number of age control points, it is better to use a linear best-fit curve.

Chronology of the Galiza, Minho and Algarve Cores The chronology of core GeoB11033-1 (Box-core of Galiza site) is based on a set of twelve 210Pb data points,

obtained in the upper 30 cm of the record, and one accelerator mass spectrometry 14C date (AMS C14), obtained in planktonic foraminifera (Table 2, Fig. 3, Fig. 4). 210Pb data was evaluated with the Constant Flux and Constant Sedimentation Rate model (CFCSR - (Appleby and Oldfield, 1992)) to date the upper 30 cm of the sediment core. The sedimentation rate was determined using the excess 210Pb (210Pbexcess) values, which is equivalent to the total 210Pb activity minus the supported 210Pb activity in equilibrium with sedimentary 226Ra. The excess 210Pb profile shows an exponential decrease with depth reaching the stable background value obtained using the 226Ra activity at 27.5 cm depth. The data points at 6 and 8 cm depth were excluded (Fig. 3). The 210Pb sedimentation rate estimated for the first 13 centimeters is 0.04 cm yr-1. Top core age was assumed to be the core recovery year, 2006. In the case of core DIVA09 GC (Minho site) the age-model construction is based in 12 210Pb data points distributed by 90 cm and 6 14C dates (AMS C14), obtained in marine material (shell and planktonic foraminifera) (Table 2, Fig. 3, Fig. 4). Background value was found at 9 cm depth. CFCSR model was defined excluding the 210Pb values at 6 cm. Top age was assumed to be the core recovery year, 2009. The 210Pb sedimentation rate estimated for the first 10 cm is 0.05 cm yr-1. The age-model of POPEI VC2B (Algarve site) is based on a set of eight 210Pb data points, obtained in the upper 50 cm of the record, and eight accelerator mass spectrometry 14C dates (AMS C14), obtained in marine material (shell and planktonic foraminifera) (Table 2, Fig. 3, Fig. 4). 210Pb data was interpreted with the CFCSR model and the data for the upper 30 cm of the sediment, as the results of two additional data points (39-40 and 49-50 cm) were negligible. The stable background value found in all the other cores was not attained, but the 210Pb estimated sedimentation rate is 0.52 cm yr-1. Top age was assumed to be the core recovery year, 2008.

Some specific comments below: Page 1 line 18: The Iberian Peninsula, at North Atlantic mid-latitude and the western extreme of the European continent, is a relevant area for climate reconstructions. – Rephrase sentence and what makes it a relevant area for climate reconstructions? The sentence was changed following reviewer 1 suggestion.

Line 25: Is that even significant as the calibration error on alkenone SST is 1.5 C? Schouten et al., 2013, http://dx.doi.org/10.1016/j.orggeochem.2012.09.006 We used the calibration method defined by (Muller et al, 1998), which is a global calibration based on core-top sediments and mean annual climatological temperatures. The error associated with this calibration was defined in the original paper: "the standard error is 1.5°C, however considering that the Uk'37 values used for the global calibration were measured in about ten laboratories which partly used different methodologies, this differences could be minor rather attesting the robustness of the Uk'37 paleotemperature indicator". Schouten et al. (2013) compiles previously published information in his Table 7. Other calibration models use suspended matter (SOM) Uk'37 calibrated to in-situ measured SST (Conte et al., 2006; Gould et al., 2017) or are based on culture data (Prahl et al., 1988) and water column measurements (Prahl et al., 2005). As a test we have used the three different models referred above to estimate SST in one of our sedimentary sequences (PO287- 6, Porto). Figure 5 shows no difference between the Muller and Prahl calibrations, while systematically lower SSTs are estimated when using Conte's calibration equation for core-top sediments (Mollenhauer et al., 2015). Independently of the used calibration method, the trends and amplitude of the observed variations are maintained all along the record even if variation is $\leq 1$ °C. As such, we conclude that our variability is significant, moreover for the definition of a long-term trend. Besides, UK'37 derived SST data has been compared to those determined from GDGTs by Mollenhauer et al. (2015) for the Mauritania upwelling system and the authors conclusion is: SST reconstructions based on alkenones are in excellent agreement with satellite data, and the entire seasonal amplitude of temperature variations at the sea surface is well recorded. In contrast, GDGT based temperature reconstructions using the logarithmic TEX86 calibration yields temperature maxima similar to observed maxima, but a reduced seasonal amplitude (warm bias).

Page 2 Line 2: change to Medieval climate anomaly Line 5: what does particular

mean? Wording has been changed

Line 7: "The intense precipitation/ flooding and warm winters but cooler intermediate seasons (spring and fall) observed for the early MWP imply the interplay of internal oceanic variability with the three atmospheric circulation modes, North Atlantic Oscillation (NAO), East Atlantic (EA) and Sandinavia (SCAND) in a positive phase".-how would the interplay of these 3 patterns cause the observed pattern? We have profoundly changed the Introduction, and these patterns are now only referred. The effect of the three modes of atmospheric circulation on the climate of the Iberia Península (shown in figure 5 of Hernández et al. (2015)) is discussed in detail in the section Climate Forcing Mechanisms of the Discussion.

Line 15: rephrase-sentence like that makes no sense Line 32: restructure Line 33: delete Medieval Warm Period (MWP) Page 3: Line 27: rephrase bad English Page 4 Line 23-26: superficial statement needs more explanation Line 30: change to: For that we combine the above mentioned published records with 3 new records located along the Iberian margin from 42_ N to 36 _N, covering the last 2,000 yr The paper was revised taking into account all of referee 1 comments and requests. Page 6: Line 4: Any additional proof that the cores are tracing river input despite pollen like BIT index We did not use the BTI index, but as stated on lines 5 to 10 of the manuscript, "Intensity of river discharge and on-land precipitation regimes were determined by using lipid compounds synthesized by higher plants, such as C23–C33 n-alkanes ([n-alc]) (e.g. Farrington et al. (1988); Pelejero et al. (1999); Prahl et al. (1994)) and the total pollen concentration (TPC)"

Page 12 Line 17-18: not clear Line 32: what does that mean important decrease? Page 13 Line 15: what does the N stands for? Line 19: rephrase Page 17 Line 20: Specific climate conditions – unclear what does specific indicate? Requested revisions were taken into consideration

List of Tables and Figures Table 1 – Results of 14C AMS dating of the gravity core

PO287-6G. Ages were reservoir corrected by 400 yr. Error column lists $\pm$ errors of 14C ages. Table 2 – Results of 14C accelerator mass spectrometry dating (means $\pm$ SE) for cores GeoB11033-1 (Galiza), DIVA09CG (Minho) and POPEI VC2B (Algarve). Ages were corrected for reservoir effect by 400 yr and converted into calendar years (AD/CE). Figure 1 Information used to construct our age model; A) 210Pb activity downcore PO287-6B; B) MS correlation of PO287-6G to sedimentary sequences from the Tagus system (Abrantes et al. 2005); C) Depth vs. AD ages (with $2\sigma$ error) for PO287-6G with a linear best fit. Figure 2 – Dated levels for core PO287-6G with a linear and a polynomial best fit for comparison. Figure 3 – 210Pb activity downcore for the box-core GeoB11033-1 at the Galiza site and cores (Minho) DIVA09GC and (Algarve) POPEI VC2B. Figure 4. Depth vs. AD ages (with $2\sigma$ error) for cores GeoB11033-1 and GC at the Galiza site (orange), DIVA09GC (Minho, magenta) and POPEI VC2B (Algarve, red), with a linear best fit. Figure 5 – Comparison of the SST variability estimated from three different calibration equations, along core PO287-6 (PORTO).

References Abrantes, F., Lebreiro, S., Rodrigues, T., Gil, I., Bartels-Jónsdóttir, H., Oliveira, P., Kissel, C., and Grimalt, J. O.: Shallow-marine sediment cores record climate variability and earthquake activity off Lisbon (Portugal) for the last 2,000 years., Quaternary Science Reviews, doi: 10.1016/j.quascirev.2004.04.009, 2005. 2005. Appleby, P. and Oldfield, F.: Applications of lead-210 to sedimentation studies. In: Uranium Series Disequelibrium. Applications to Earth, Marine and Environmental Sciences., Ivanovich, M. and Harmon, M. (Eds.), Clarendon Press, Oxford, 1992. Conte, M. H., Sicre, M.-A., Rühlemann, C., Weber, J. C., Schulte, S., Schulz-Bull, D., and Blanz, T.: Global temperature calibration of the alkenone unsaturation index (UKâǍš37) in surface waters and comparison with surface sediments, Geochemistry, Geophysics, Geosystems, 7, n/a-n/a, 2006. Farrington, J. W., Davis A. C., Sulanowski J., McCaffrey M. A., McCarthy M., Clifford C. H., P., D., and K., V. J.: Biogeochemistry of lipids in surface sediments of the Peru Upwelling Area at 15°S. , Org. Geochem. , 13, 607-617, 1988. Gould, J., Kienast, M., and Dowd, M.: Investigation of the UK37' vs. SST relationship for Atlantic Ocean suspended particulate alkenones: An

alternative regression model and discussion of possible sampling bias, Deep-Sea Research Part I, 123, 13-21, 2017. Hernández, A., Trigo, R. M., Pla-Rabes, S., Valero-Garcés, B. L., Jerez, S., Rico-Herrero, M., Vega, J. C., Jambrina-Enríquez, M., and Giralt, S.: Sensitivity of two Iberian lakes to North Atlantic atmospheric circulation modes, Climate Dynamics, 45, 3403-3417, 2015. Mollenhauer, G., Basse, A., Kim, J.-H., Sinninghe Damsté, J. S., and Fischer, G.: A four-year record of UKâǍš37- and TEX86-derived sea surface temperature estimates from sinking particles in the filamentous upwelling region off Cape Blanc, Mauritania, Deep Sea Research Part I: Oceanographic Research Papers, 97, 67-79, 2015. Narayan, N., Paul, A., Mulitza, S., and Schulz, M.: Trends in coastal upwelling intensity during the late 20th century, Ocean Sci., 6, 815-823, 2010. Pelejero, C., Keinast, M., Wang, L., and Grimalt, J. O.: The flooding of Sundaland during the last deglaciation: imprints in hemipelagic sediments from the southern South China Sea, Earth and Planetary Science Letters, 171, 661-671, 1999. Prahl, F. G., Ertel, J. R., Goni, M. A., Sparrow, M. A., and Eversmeyer, B.: Terrestrial organic carbon contributions to sediments on the Washington margin., Geoch. Cosmochim. Acta, 1994. 1994. Prahl, F. G., Muehlhausen, L. A., and Zahnle, D. L.: Further evaluation of long-chain alkenones as indicators of paleoceanographic conditions, Geochimica et Cosmochimica Acta, 52, 2303-2310, 1988. Prahl, F. G., Popp, B. N., Karl, D. M., and Sparrow, M. A.: Ecology and biogeochemistry of alkenone production at Station ALOHA, Deep Sea Research Part I: Oceanographic Research Papers, 52, 699-719, 2005. Reimer, P., Baillie, M., Bard, E., Bayliss, A., Beck, J., Bertrand, C., Blackwell, P., Buck, C., Burr, G., Cutler, K., Damon, P., Edwards, R., Fairbanks, R., Friedrich, M., Guilderson, T., Hughen, K., Kromer, B., McCormac, F., Manning, Ramsey, C. B., Reimer, R., Remmele, S., Southon, J., Stuiver, M., Talamo, S., Taylor, F., Plicht, J. v. d., and Weyhenmeyer, C.: Marine04 Marine radiocarbon age calibration, 26 - 0 ka BP, Radiocarbon, 46, 1029-1058, 2004. Schouten, S., Hopmans, E. C., and Sinninghe Damsté, J. S.: The organic geochemistry of glycerol dialkyl glycerol tetraether lipids: A review, Organic Geochemistry, 54, 19-61, 2013.

Please also note the supplement to this comment:
https://www.clim-past-discuss.net/cp-2017-84/cp-2017-84-AC1-supplement.pdf

[Figure]

Figure 1 Information used to construct our age model; A) ²¹⁰Pb activity downcore PO287-6B; B) MS correlation of PO287-6G to sedimentary sequences from the Tagus system (Abrantes *et al.* 2005); C) Depth *vs.* AD ages (with 2σ error) for PO287-6G with a linear best fit.

Fig. 1.

[Figure]

**Figure 2 – Dated levels for core PO287-6G with a linear and a polynomial best fit for comparison.**

[Figure]

**Figure 3 – ²¹⁰Pb activity downcore for the box-core GeoB11033-1 at the Galiza site and cores (Minho) POPEI VC2B.**

[Figure]

**Figure 4. Depth *vs*. AD ages (with 2σ error) for cores GeoB11033-1 and GC at the Galiza site (or magenta) and POPEI VC2B (Algarve, red), with a linear best fit.**

[Figure]

**Figure 5 – Comparison of the SST variability estimated from three different calibration equations, alon**

| Sample ID | Depth (cm) | C14 Age (RC = 400 yr) | Error | Age AD | Description |
|---|---|---|---|---|---|
| KIA 35149 | 100.5 | 160 | 25 | 1770 | mixed benthics |
| KIA 29290 | 318.0 | 405 | 35 | 1478 | mixed planktonics |
| KIA 35150 | 400.0 | 820 | 30 | 1223 | mixed benthics |

**Table 1 – Results of [14]C AMS dating of the gravity core PO287-6G. Ages were reservoir corrected by 4 errors of 14C ages.**

| Core ID and depth (cm) | Laboratory code | Sample Type | Conventional [14]C age (BP) | error | Calibrated age ranges at 95% confidence intervals | Age AD |
|---|---|---|---|---|---|---|
| **GeoB11033-1** | | | | | | |
| 27 - 28.5 | OS-97151 | Foraminifera | 2430 | 25 | 746-530 | -638 |
| **DIVA 09GC** | | | | | | |
| 3 - 4 | KIA 42919 | Mollusk shell | 465 | 25 | 1841-1859 | 1864 |
| 48-49 | OS-97148 | Foraminifera | 1270 | 25 | 1057-1211 | 1133 |
| 57-58 | KIA 42920 | Mollusk shell | 1730 | 30 | 602-728 | 660 |
| 68-69 | OS-97149 | Foraminifera | 1990 | 25 | 298-482 | 400 |
| 83 - 84 | KIA 42921 | Mollusk shell | 2380 | 30 | -157 -33 | -60 |
| 101 - 102 | KIA 42922 | Mollusk shell | 2325 | 30 | -87 - 95 | 11 |
| POPEI VC2B | | | | | | |
| 130.9 | Beta 278216 | Mollusk shell | 1220 | 40 | 1080:1274 | 1184 |
| 200.6 | OS-97152 | Foraminifera | 2130 | 25 | 146:326 | 233 |
| 270.3 | OS-97143 | Foraminifera | 3020 | 25 | -902:-783 | -837 |

**Table 2 – Results of [14]C accelerator mass spectrometry dating (means ± SE) for cores GeoB11033-1 (G and POPEI VC2B (Algarve). Ages were corrected for reservoir effect by 400 yr and converted into cale**

**Supplement:**

**The Climate of the Common Era off the Iberian Peninsula**

**Response to Anonymous Referee #1**

Abrantes[1,2], Fátima; Teresa Rodrigues[1,2], Marta Rufino[2,3]; Emília Salgueiro[1,2]; Dulce Oliveira[1,2,4]; Sandra Gomes[1]; Paulo Oliveira[1]; Ana Costa[5]; Mário Mil-Homens[1]; Teresa Drago[1,6]; Filipa Naughton[1,2];

5    1 –Portuguese Institute for the Ocean and Atmosphere (IPMA), Divisão de Geologia Marinha (DivGM), Rua Alferedo Magalhães Ramalho 6, Lisboa, Portugal

2 - CCMAR, Centro de Ciências do Mar, Universidade do Algarve, Campus de Gambelas, 8005-139 Faro, Portugal

3 – IFREMER - Centre Atlantique (French Research Institute for Exploitation of the Sea, Département Ecologie et Modèles pour l'Halieutique (EMH), Rue de l'Ile d'Yeu - BP 21105, 44311 Nantes cedex 3, France

10    4 – Univ. Bordeaux, EPOC, UMR 5805, F-33615 Pessac, France

5 - Centro de Investigação em Biodiversidade e Recursos Genéticos (EnvArchCIBIO/InBIO) and Archaeosciences Laboratory (LARC/DGPC), Rua da Bica do Marquês, 2, 1300-087, Lisboa, Portugal

6 – Instituto Dom Luiz, Universidade de Lisboa, 1749-016 Lisboa, Portugal

*Correspondence to*: Fatima Abrantes (Fatima.abrantes@ipma.pt)

15    The authors thank referee 1 for his important contribution to improve the manuscript.

In general, the reviewer considers that the presented records provide useful information about climate variability offshore the Iberian Peninsula over the Common Era. However, the reviewer finds the paper too long and unable to pass a clear message, and suggests the paper to concentrate on answering a clear question.

The length of the paper has been substantially reduced. The introduction was shortened, the material and methods was 20  reduced to the essential information and most of the detailed and considered important information is now compiled as Supplementary material. The essential information relative to the cores chronology was included in the methods and the individualized chronology section of the previous version was deleted. Detailed information on the age-model construction for the new sedimentary sequences is now also included in the supplementary material. The results and discussion section was subdivided and results are now presented separately. The discussion has also been re-organized around the specific 25  questions raised by the data. Abstract and Conclusions have equally been re-written in what we hope to be a more concise style.

We certainly hope that the re-organization of the paper makes it easier to read and helps to better convey the message(s) included.

**The reviewer considers also that the paper should definitely be proof read by a native English speaker, as many parts 30  of the paper are very hard to understand lacking a sentence structure and words.**

The new and much changed version has been thoroughly revised by a native English speaker.

**Age model: The 3 new age models of the cores should be shown as an age-depth plot additionally to the table with the 14C dates. Moreover, a Bayesian age depth model should be performed to better constrain age uncertainties.**

An explanation and data used for the definition of the age-models for the new three cores is now included in the 35  supplementary material. However, in order to correctly respond to this comment, below we present a discussion on the methodology used for the age model construction of all 7 sedimentary sequences used in this paper, the comparison between methods and the basic data for the three new sedimentary sequences, including de age-depth models.

Chronology: The example of PO287-6

The age-model for the spliced sequence composed of cores PO287-6B and 6G (box, gravity) was constructed by combining two methods: (1) $^{210}$Pb activity measured in box-core samples (Fig. 1A) which depending on the accepted model provides a sedimentation rate varying between 0.32 and 0.43 cm yr$^{-1}$; (2) four accelerator mass spectrometry (AMS) radiocarbon measurements (Leibniz-Laboratory for Radiometric Dating and Stable Isotope Research, Kiel, Germany) (Table 1). Two further ages were assigned through MS correlation to other well-dated cores recovered off Lisbon (Fig. 1B). Raw AMS $^{14}$C dates were corrected for reservoir effect by 400 yr (Abrantes et al., 2005) and converted to calendar ages with the INTCAL04 data set (Reimer et al., 2004). The obtained calendar ages are presented in years Anno Domini (AD/CE).

To develop a continuous record, the splicing of the long cores (piston and gravity) with the box-core (PO287-6B, 6G) was done through the Magnetic Susceptibility record (MS) of both cores (Fig. 1B). Further integration of the above-referred cores was based on the 1952 CE age found at 20.7 cm (depth corrected for compaction during sub-sampling) in box-core PO287-6B. Comparison of the PO287-6G MS record to sedimentary sequences from the Tagus system (Abrantes *et al*. 2005) was also done; Figure 1C depicts Depth *vs*. AD ages (with 2σ error) for PO287-6G with a linear best fit. An age that is within the error of the age estimated for the same depth using the sedimentation rate that results from a linear interpolation of the five considered levels (Table 1, Figs. 1B, 1C).

Given the uncertainty associated to the 14C dates, the establishment of an age model based on the interpolation between each dated level is normally avoided for sequences covering short time intervals (Jan Heinemeier, pers com.). An age/depth relationship defined by the linear best-fit line of the calibrated $^{14}$C ages is the most common approach (e.g. (Narayan et al., 2010)). However we decided to compare age-depth models using both a linear and a polynomial best fit for core PO287-6G (Fig. 2). Both models give very close ages on the interval with dated levels, but the lack of dates at the base of core PO287-6G leads to older ages at the bottom of the record when using the polynomial solution.

| Sample ID | Depth (cm) | C14 Age (RC = 400 yr) | Error | Age AD | Description |
|-----------|-----------|-----------------------|-------|--------|-------------|
| KIA 35149 | 100.5 | 160 | 25 | 1770 | mixed benthics |
| KIA 29290 | 318.0 | 405 | 35 | 1478 | mixed planktonics |
| KIA 35150 | 400.0 | 820 | 30 | 1223 | mixed benthics |

**Table 1 – Results of $^{14}$C AMS dating of the gravity core PO287-6G. Ages were reservoir corrected by 400 yr. Error column lists ± errors of 14C ages.**

[Figure]

[Figure]

Figure 1 Information used to construct our age model; A) [210]Pb activity downcore PO287-6B; B) MS correlation of PO287-6G to sedimentary sequences from the Tagus system (Abrantes *et al.* 2005); C) Depth *vs.* AD ages (with 2σ error) for PO287-6G with a linear best fit.

[Figure]

Figure 2 – Dated levels for core PO287-6G with a linear and a polynomial best fit for comparison.

**Why the selection of a linear interpolation?**

The assumption of a constant sedimentation rate was applied in Abrantes et al., 2005 (QSR) following the advise of Jan

Heinemeier (Aharus University [14]C dating center). According to this expert, in the case of records covering short time-scales, such as the last 2,000 yr, and with a relatively small number of age control points, it is better to use a linear best-fit curve.

Chronology of the Galiza, Minho and Algarve Cores

[Figure]

**Figure 3 – $^{210}$Pb activity downcore for the box-core GeoB11033-1 at the Galiza site and cores (Minho) DIVA09GC and (Algarve) POPEI VC2B.**

The chronology of core GeoB11033-1 (Box-core of Galiza site) is based on a set of twelve $^{210}$Pb data points, obtained in the
10  upper 30 cm of the record, and one accelerator mass spectrometry $^{14}$C date (AMS C14), obtained in planktonic foraminifera (Table 2, Fig. 3, Fig. 4).

| Core ID and depth (cm) | Laboratory code | Sample Type | Conventional $^{14}$C age (BP) | error | Calibrated age ranges at 95% confidence intervals | Age AD | Laboratory |
|---|---|---|---|---|---|---|---|
| **GeoB11033-1** | | | | | | | |
| 27 - 28.5 | OS-97151 | Foraminifera | 2430 | 25 | 746-530 | -638 | National Ocean Sciences AMS - WHOI |
| **DIVA 09GC** | | | | | | | |
| 3 - 4 | KIA 42919 | Mollusk shell | 465 | 25 | 1841-1859 | 1864 | Leibniz Labor - Kiel |
| 48-49 | OS-97148 | Foraminifera | 1270 | 25 | 1057-1211 | 1133 | National Ocean Sciences AMS - WHOI |
| 57-58 | KIA 42920 | Mollusk shell | 1730 | 30 | 602-728 | 660 | Leibniz Labor - Kiel |
| 68-69 | OS-97149 | Foraminifera | 1990 | 25 | 298-482 | 400 | National Ocean Sciences AMS - WHOI |
| 83 - 84 | KIA 42921 | Mollusk shell | 2380 | 30 | -157 -33 | -60 | Leibniz Labor - Kiel |
| 101 - 102 | KIA 42922 | Mollusk shell | 2325 | 30 | -87 - 95 | 11 | Leibniz Labor - Kiel |
| **POPEI VC2B** | | | | | | | |
| 130.9 | Beta 278216 | Mollusk shell | 1220 | 40 | 1080:1274 | 1184 | Beta Analytics |
| 200.6 | OS-97152 | Foraminifera | 2130 | 25 | 146:326 | 233 | National Ocean Sciences AMS - WHOI |
| 270.3 | OS-97143 | Foraminifera | 3020 | 25 | -902:-783 | -837 | National Ocean Sciences AMS - WHOI |

**Table 2 – Results of $^{14}$C accelerator mass spectrometry dating (means ± SE) for cores GeoB11033-1 (Galiza), DIVA09CG (Minho) and POPEI VC2B (Algarve). Ages were corrected for reservoir effect by 400 yr and converted into calendar years (AD/CE).**

15  $^{210}$Pb data was evaluated with the Constant Flux and Constant Sedimentation Rate model (CFCSR - (Appleby and Oldfield, 1992)) to date the upper 30 cm of the sediment core. The sedimentation rate was determined using the excess $^{210}$Pb ($^{210}$Pb$_{excess}$) values, which is equivalent to the total $^{210}$Pb activity minus the supported $^{210}$Pb activity in equilibrium with sedimentary $^{226}$Ra. The excess $^{210}$Pb profile shows an exponential decrease with depth reaching the stable background value obtained using the $^{226}$Ra activity at 27.5 cm depth. The data points at 6 and 8 cm depth were excluded (Fig. 3). The $^{210}$Pb
20  sedimentation rate estimated for the first 13 centimeters is 0.04 cm yr$^{-1}$. Top core age was assumed to be the core recovery year, 2006.

In the case of core DIVA09 GC (Minho site) the age-model construction is based in 12 $^{210}$Pb data points distributed by 90 cm and 6 $^{14}$C dates (AMS C14), obtained in marine material (shell and planktonic foraminifera) (Table 2, Fig. 3, Fig. 4). Background value was found at 9 cm depth. CFCSR model was defined excluding the $^{210}$Pb values at 6 cm. Top age was
25  assumed to be the core recovery year, 2009. The $^{210}$Pb sedimentation rate estimated for the first 10 cm is 0.05 cm yr$^{-1}$.

The age-model of POPEI VC2B (Algarve site) is based on a set of eight $^{210}$Pb data points, obtained in the upper 50 cm of the record, and eight accelerator mass spectrometry $^{14}$C dates (AMS C14), obtained in marine material (shell and planktonic

foraminifera) (Table 2, Fig. 3, Fig. 4).

[210]Pb data was interpreted with the CFCSR model and the data for the upper 30 cm of the sediment, as the results of two additional data points (39-40 and 49-50 cm) were negligible. The stable background value found in all the other cores was not attained, but the [210]Pb estimated sedimentation rate is 0.52 cm yr[-1]. Top age was assumed to be the core recovery year,

5    2008.

[Figure]

**Figure 4. Depth *vs.* AD ages (with 2σ error) for cores GeoB11033-1 and GC at the Galiza site (orange), DIVA09GC (Minho, magenta) and POPEI VC2B (Algarve, red), with a linear best fit.**

10    **Some specific comments below:**

**Page 1 line 18: The Iberian Peninsula, at North Atlantic mid-latitude and the western extreme of the European continent, is a relevant area for climate reconstructions. – Rephrase sentence and what makes it a relevant area for climate reconstructions?**

The sentence was changed following reviewer 1 suggestion.

15    **Line 25: Is that even significant as the calibration error on alkenone SST is 1.5 C? Schouten et al., 2013, http://dx.doi.org/10.1016/j.orggeochem.2012.09.006**

We used the calibration method defined by (Muller et al, 1998), which is a global calibration based on core-top sediments and mean annual climatological temperatures. The error associated with this calibration was defined in the original paper: "*the standard error is 1.5ºC, however considering that the Uk'37 values used for the global calibration were measured in*

20    *about ten laboratories which partly used different methodologies, this differences could be minor rather attesting the robustness of the Uk'37 paleotemperature indicator*". Schouten et al. (2013) compiles previously published information in his Table 7.

Other calibration models use suspended matter (SOM) Uk'37 calibrated to in-situ measured SST (Conte et al., 2006; Gould et al., 2017) or are based on culture data (Prahl et al., 1988) and water column measurements (Prahl et al., 2005).

25    As a test we have used the three different models referred above to estimate SST in one of our sedimentary sequences (PO287- 6, Porto). Figure 5 shows no difference between the Muller and Prahl calibrations, while systematically lower SSTs are estimated when using Conte's calibration equation for core-top sediments (Mollenhauer et al., 2015). Independently of the used calibration method, the trends and amplitude of the observed variations are maintained all along the record even if variation is ≤ 1 ºC. As such, we conclude that our variability is significant, moreover for the definition of a long-term trend.

[Figure]

**Figure 5 – Comparison of the SST variability estimated from three different calibration equations, along core PO287-6 (PORTO).**

Besides, UK'37 derived SST data has been compared to those determined from GDGTs by Mollenhauer et al. (2015) for the Mauritania upwelling system and the authors conclusion is: *SST reconstructions based on alkenones are in excellent agreement with satellite data, and the entire seasonal amplitude of temperature variations at the sea surface is well recorded. In contrast, GDGT based temperature reconstructions using the logarithmic TEX86 calibration yields temperature maxima similar to observed maxima, but a reduced seasonal amplitude (warm bias).*

**Page 2 Line 2: change to Medieval climate anomaly**
**Line 5: what does particular mean?**

Wording has been changed

**Line 7: "The intense precipitation/ flooding and warm winters but cooler intermediate seasons (spring and fall) observed for the early MWP imply the interplay of internal oceanic variability with the three atmospheric circulation modes, North Atlantic Oscillation (NAO), East Atlantic (EA) and Sandinavia (SCAND) in a positive phase".-how would the interplay of these 3 patterns cause the observed pattern?**

We have profoundly changed the Introduction, and these patterns are now only referred. The effect of the three modes of atmospheric circulation on the climate of the Iberia Península (shown in figure 5 of Hernández et al. (2015)) is discussed in detail in the section Climate Forcing Mechanisms of the Discussion.

**Line 15: rephrase-sentence like that makes no sense**
**Line 32: restructure**
**Line 33: delete Medieval Warm Period (MWP)**
**Page 3: Line 27: rephrase bad English**
**Page 4 Line 23-26: superficial statement needs more explanation**
**Line 30: change to: For that we combine the above mentioned published records with 3 new records located along the Iberian margin from 42_ N to 36 _N, covering the last 2,000 yr**

The paper was revised taking into account all of referee 1 comments and requests.

**Page 6: Line 4: Any additional proof that the cores are tracing river input despite pollen like BIT index**

We did not use the BTI index, but as stated on lines 5 to 10 of the manuscript, "*Intensity of river discharge and on-land precipitation regimes were determined by using lipid compounds synthesized by higher plants, such as C23–C33 n-alkanes ([n-alc]) (e.g. Farrington et al. (1988); Pelejero et al. (1999); Prahl et al. (1994)) and the total pollen concentration (TPC)*"

**Page 12 Line 17-18: not clear**

**Line 32: what does that mean important decrease?**
**Page 13 Line 15: what does the N stands for? Line 19: rephrase**
**Page 17 Line 20: Specific climate conditions – unclear what does specific indicate?**

Requested revisions were taken into consideration

5 **References**

Abrantes, F., Lebreiro, S., Rodrigues, T., Gil, I., Bartels-Jónsdóttir, H., Oliveira, P., Kissel, C., and Grimalt, J. O.: Shallow-marine sediment cores record climate variability and earthquake activity off Lisbon (Portugal) for the last 2,000 years., Quaternary Science Reviews, doi: 10.1016/j.quascirev.2004.04.009, 2005. 2005.

Appleby, P. and Oldfield, F.: Applications of lead-210 to sedimentation studies. In: Uranium Series Disequelibrium.

10 Applications to Earth, Marine and Environmental Sciences., Ivanovich, M. and Harmon, M. (Eds.), Clarendon Press, Oxford, 1992.

Conte, M. H., Sicre, M.-A., Rühlemann, C., Weber, J. C., Schulte, S., Schulz-Bull, D., and Blanz, T.: Global temperature calibration of the alkenone unsaturation index (UK′37) in surface waters and comparison with surface sediments, Geochemistry, Geophysics, Geosystems, 7, n/a-n/a, 2006.

15 Farrington, J. W., Davis A. C., Sulanowski J., McCaffrey M. A., McCarthy M., Clifford C. H., P., D., and K., V. J.: Biogeochemistry of lipids in surface sediments of the Peru Upwelling Area at 15°S. , Org. Geochem. , 13, 607-617, 1988.

Gould, J., Kienast, M., and Dowd, M.: Investigation of the UK37' vs. SST relationship for Atlantic Ocean suspended particulate alkenones: An alternative regression model and discussion of possible sampling bias, Deep-Sea Research Part I, 123, 13-21, 2017.

20 Hernández, A., Trigo, R. M., Pla-Rabes, S., Valero-Garcés, B. L., Jerez, S., Rico-Herrero, M., Vega, J. C., Jambrina-Enríquez, M., and Giralt, S.: Sensitivity of two Iberian lakes to North Atlantic atmospheric circulation modes, Climate Dynamics, 45, 3403-3417, 2015.

Mollenhauer, G., Basse, A., Kim, J.-H., Sinninghe Damsté, J. S., and Fischer, G.: A four-year record of UK′37- and TEX86-derived sea surface temperature estimates from sinking particles in the filamentous upwelling region off Cape Blanc,

25 Mauritania, Deep Sea Research Part I: Oceanographic Research Papers, 97, 67-79, 2015.

Narayan, N., Paul, A., Mulitza, S., and Schulz, M.: Trends in coastal upwelling intensity during the late 20th century, Ocean Sci., 6, 815-823, 2010.

Pelejero, C., Keinast, M., Wang, L., and Grimalt, J. O.: The flooding of Sundaland during the last deglaciation: imprints in hemipelagic sediments from the southern South China Sea, Earth and Planetary Science Letters, 171, 661-671, 1999.

30 Prahl, F. G., Ertel, J. R., Goni, M. A., Sparrow, M. A., and Eversmeyer, B.: Terrestrial organic carbon contributions to sediments on the Washington margin., Geoch. Cosmochim. Acta, 1994. 1994.

Prahl, F. G., Muehlhausen, L. A., and Zahnle, D. L.: Further evaluation of long-chain alkenones as indicators of paleoceanographic conditions, Geochimica et Cosmochimica Acta, 52, 2303-2310, 1988.

Prahl, F. G., Popp, B. N., Karl, D. M., and Sparrow, M. A.: Ecology and biogeochemistry of alkenone production at Station

35 ALOHA, Deep Sea Research Part I: Oceanographic Research Papers, 52, 699-719, 2005.

Reimer, P., Baillie, M., Bard, E., Bayliss, A., Beck, J., Bertrand, C., Blackwell, P., Buck, C., Burr, G., Cutler, K., Damon, P., Edwards, R., Fairbanks, R., Friedrich, M., Guilderson, T., Hughen, K., Kromer, B., McCormac, F., Manning, Ramsey, C. B., Reimer, R., Remmele, S., Southon, J., Stuiver, M., Talamo, S., Taylor, F., Plicht, J. v. d., and Weyhenmeyer, C.: Marine04 Marine radiocarbon age calibration, 26 - 0 ka BP, Radiocarbon, 46, 1029-1058, 2004.

40 Schouten, S., Hopmans, E. C., and Sinninghe Damsté, J. S.: The organic geochemistry of glycerol dialkyl glycerol tetraether lipids: A review, Organic Geochemistry, 54, 19-61, 2013.

**The Climate of the Common Era off the Iberian Peninsula**

Abrantes[1,2], Fátima; Teresa Rodrigues[1,2], Marta Rufino[2,3]; Emília Salgueiro[1,2]; Dulce Oliveira[1,2,4]; Sandra Gomes[1]; Paulo Oliveira[1]; Ana Costa[5]; Mário Mil-Homens[1]; Teresa Drago[1, 6]; Filipa Naughton[1,2,]

1 – Portuguese Institute for Sea and Atmosphere (IPMA), Divisão de Geologia Marinha (DivGM), Rua Alferedo Magalhães Ramalho 6, Lisboa, Portugal

2 - CCMAR, Centro de Ciências do Mar, Universidade do Algarve, Campus de Gambelas, 8005-139 Faro, Portugal

3 – IFREMER - Centre Atlantique (French Research Institute for Exploitation of the Sea, Département Ecologie et Modèles pour l'Halieutique (EMH), Rue de l'Ile d'Yeu - BP 21105, 44311 Nantes cedex 3, France

4 – Univ. Bordeaux, EPOC, UMR 5805, F-33615 Pessac, France

5 - Centro de Investigação em Biodiversidade e Recursos Genéticos (EnvArchCIBIO/InBIO) and Archaeosciences Laboratory (LARC/DGPC), Rua da Bica do Marquês, 2, 1300-087, Lisboa, Portugal

6 – Instituto Dom Luiz, Universidade de Lisboa, 1749-016 Lisboa, Portugal

*Correspondence to*: Fatima Abrantes (Fatima.abrantes@ipma.pt)

Key Words – Last 2,000 yr, climate, SST, precipitation, Iberian Peninsula

**Abstract.** The Mediterranean region is a climate hot spot, sensitive not only to global warming but also to water availability. In this work we document major temperature and precipitation changes in the Iberian Peninsula during the last 2,000 yr, and propose an interplay of the North Atlantic internal variability with the three modes of atmospheric circulation (North Atlantic Oscillation (NAO), East Atlantic (EA) and Scandinavia (SCAND)) to explain the observed climate variability.

Reconstructions of Sea Surface Temperature (SST derived from alkenones) and on-land precipitation (estimated from higher plant n-alkanes and pollen data) in sedimentary sequences recovered at 5 sites along the Iberian Margin between the South of Portugal (Algarve) and the Northwest of Spain (Galiza) (36 to 42 ºN) constitute our database.

A clear long-term cooling trend up to the beginning of the 20th century emerges in all SST records and is considered a reflection of the decrease in the Northern Hemisphere summer insolation that began in the Holocene optimum. Multi-decadal/ centennial SST variability follows other records from Spain, Europe and the Northern Hemisphere. Warm SSTs throughout the first 1300 yr encompass the Roman Period (RP), the Dark Ages (DA) and the Medieval Climate Anomaly (MCA). A cooling initiated at 1300 CE, leads to 4 centuries of colder SSTs contemporary with the Little Ice Age (LIA). the Industrial Era is marked by climate warming since 1800 CE..

Novel results include two distinct phases in the MCA, an early period (900 – 1100 yr) characterized by intense precipitation/ flooding and warm winters but a cooler spring-fall season attributed to the interplay of internal oceanic variability with the three positive modes of atmospheric circulation (NAO, EA and SCAND). The late MCA is marked by cooler and relatively drier winters and a warmer spring-fall season consistent with a change in of the SCAND to a negative mode.

The beginning of the Industrial Era is coherent with a stronger impact of internal oceanic variability on the climate of the Western Iberian Peninsula. A particularly noticeable rise in SST at the Algarve site by mid 20th century (± 1970) is proposed to be a reflection of the expected regional response to the ongoing climate warming.

---

## Author Comment (AC2) · 12 Nov 2017

Please consider reading the supplement for a more clear view of this response

This reviewer seconds the general opinion and concerns of reviewer 1 and prefers to present specific comments. We thank this valuable review and to not repeat ourselves, we invite the reviewer to please read our response to reviewer1. Specific comments are addressed below.

Specific comments: -page 1: Abstract, first sentence, is not convincing at all. Please remove it. -page 2: Lines 6-11, those lines are too complex and could not be properly understood without having a read over the modern climatology chapter. -page 2: Line 23, perhaps cite Guiot and Cramer, 2016, Science, for a more recent assessment. The

[Figure]

new version of the paper takes into account all of Referee 2 requests and comments.

-page 3: Here the discussion would greatly benefit if the authors could add a series of very simple figures introducing the NAO, EA and SCAND modes of climate variability, in particular since the authors often refer to those modes later in the discussion. In order to reduce the size and focus of the introductory text, the explanations and discussion of the impact of these modes of atmospheric circulation on the climate of the Iberia Peninsula is now only considered in the discussion section. However, to clarify this important aspect, we have used the maps that show the regional effect of all three modes of atmospheric circulation (NAO, EA and SCAND) on the SST and precipitation for both winter and summer conditions over the Iberian Peninsula (Fig. 1), as proposed by Hernández et al., (2015) and presented in their figure 5 (Hernández et al., 2015). However, this work is published and we can only refer to the figure

-page 4: Here the reader is really lost, and could not remember any clear information at the end of the page. The Introduction was fully rewritten and reduced in length.

-page 6: Line 31, please check that ''standardized'' and ''scaled'' are not referred to ''normalized'' and ''standardized'' instead. We agree with the referee that there is some confusion over these terms, which are often interchanged. This is why for the purpose of clarity, we have added under brackets the mathematic operations that were actually carried out. Normalizing typically means to transform the observations such that they look normally distributed <http://en.wikipedia.org/wiki/Normal_distribution>. Some examples of transformations for normalizing data are power transformations <http://en.wikipedia.org/wiki/Power_transform> (e.g. log). Scaling simply means multiplying your observations by a constant c, which changes the scale (for example from nanometers to kilometers). Scaling is generally done for convenience, and does not imply any change in the distribution of the variable. Standardizing generally means subtracting the mean and dividing by the standard deviation. But there often the terms are interchanged through the processes, i.e. scaled is named when we normalized, etc. the concepts are nested within each other. For example, the function 'scale' in

R performs what is often named as standardizing the variables in a PCA, which corresponds to centering + scaling. Therefore, normalizing would be more transforming into a normal variable, which according to the bibliography it would not be applicable in this case. Either centering, scaling or standardizing would be ok for us if the referee considers so.

-page 8: Line 2, ''All age models : : : all accepted 14C dated levels" reads like you've discarded some of them. Please clarify the age model description. The chronology for the three new cores will be included as Supplementary material. A presentation of the used data and methods is included in the response to Reviewer1. Given that these comments are available on line we decided by not repeating that information and respectfully ask Reviewer 2 to please read that response to Reviewer 1.

-page 9: Lines 23-31, the discussion on the most recent SST shifts could be either discussed later, or more developed (what is the great salinity minimum?). It is difficult to see what happens over the last 50 years. We agree with reviewer 1 suggestion and a more detailed discussion on the SST variability within the Industrial Era section is now presented in the last point of the discussion of this new version of the paper.

-page 10: on the n-alcane concentrations, lines 1-10 please explain more how you calibrate the proxy. I would intuitively expect that dilution plays an important role, so that the more riverine runoff you get, the more alcanes would be diluted by terrigenous material, but it seems to be the contrary: : : The more terrigenous material the higher the [n-alc], or is it diluted by the terrigenous component? n-alkanes ([n-alk]) are long linear chain lipid molecules that mostly originate from cuticles of the vascular plants, and their concentration in oceanic sediments has been widely used as a proxy for river discharge (e.g. ((Elias et al., 1997; Grimalt et al., 1990)). Furthermore, previous work on Iberian Margin has shown a good agreement between [n-alk] and River flux (Abrantes et al., 2005; Rodrigues et al., 2009). The assessment of the value of this proxy at the regional scale, now included in the supplementary material, was done through the comparison of the [n-alk] data obtained for the most recent sediments of the Porto, Tejo and Algarve

sites with the average river runoff for the Douro, Tejo and Guadiana Rivers during the NAO winter months (DJFM) for the years after 1991 and available at the Portuguese National Service for Hidric Resources (SNIRH) (http://snirh.inag.pt). The results reveal a significant (at p>0.01) Pearson correlation of 0.54 and n=47, confirming [n-alk] as a good proxy for evaluating the intensity of River runoff on the Iberian Peninsula. Most of the sediments in these depocenters are muds (silt and clay) that result from the deposition of fine particles of terrigenous origin that are transported into the ocean in suspension by the river plumes (e.g (Abrantes et al., 2005; Abrantes et al., 2011)). Furthermore, most of the organic matter is bonded to the fine fraction of the sediment, in particular the clay fraction (Mil-Homens et al., 2007). The high correlation of the [n-alk] to other proxies of continental origin, such as Fe, has been demonstrated in previous papers for the Tejo area (Abrantes et al., 2005; Rodrigues et al., 2009). If we consider the sediments Fe content (cps) not only for the Tejo area but also for the Porto site and compare it to the n-alkanes measured concentrations in the same cores, a significant (at p>0.01) Pearson correlation of 0.47 and n=250 is found, revealing a parallel increase on both components of continental origin.

-chapter 5.3: please try to be more concise through sorting out the results and discussion separately. Why are you not discussing the LIA? Also, I find the wavelet analysis neither convincing nor useful to the discussion, and I don't see how you could extract significant periodicities longer than a century over time windows shorter than two centuries. We do refer and discussed the LIA pattern found in our records, however, the fact that they can be explained by previously highly discussed and published climate processes lead us to concentrate on the discussion of the periods that reveal marked differences, the MWP and the last 500 years.

-figures: please check the captions. In general, there are too many panels. We have looked into this aspect in detail, but still feel the need to maintain most of the panels. However, we have been very careful in referring to the figure's panel identification, every time one of them is mentioned in the text. We hope that this has made it easier

to read previous figures, 5, 6 and 6 and will in the new version be figures 2, 3 and 4.

List of Figures

References Abrantes, F., Lebreiro, S., Rodrigues, T., Gil, I., Bartels-Jónsdóttir, H., Oliveira, P., Kissel, C., and Grimalt, J. O.: Shallow-marine sediment cores record climate variability and earthquake activity off Lisbon (Portugal) for the last 2,000 years., Quaternary Science Reviews, doi: 10.1016/j.quascirev.2004.04.009, 2005. 2005. Abrantes, F., Rodrigues, T., Montanari, B., Santos, C., Witt, L., Lopes, C., and Voelker, A. H. L.: Climate of the last millennium at the southern pole of the North Atlantic Oscillation: an inner-shelf sediment record of flooding and upwelling, Climate Research, 48, 261-280, 2011. Elias, V., Simoneit, B., and Cardoso, J. N.: Even N-Alkane Predominances on the Amazon Shelf and A Northeast Pacific Hydrothermal System, J. Naturwissenschaften 84, 1997. Grimalt, J. O., P. Fernández, J. Bayona, and Albaige′s, J.: Assessment of fecal sterols and ketones as indicators of urban sewage input to coastal waters,, Environ. Sci. Technol., 24, 357-363, 1990. Hernández, A., Trigo, R. M., Pla-Rabes, S., Valero-Garcés, B. L., Jerez, S., Rico-Herrero, M., Vega, J. C., Jambrina-Enríquez, M., and Giralt, S.: Sensitivity of two Iberian lakes to North Atlantic atmospheric circulation modes, Climate Dynamics, 45, 3403-3417, 2015. Mil-Homens, M., Stevens, R. L., Cato, I., and Abrantes, F.: Regional geochemical baselines for Portuguese shelf sediments, Environmental Pollution, 148, 418-427, 2007. Rodrigues, T., Grimalt, J. O., Abrantes, F. G., Flores, J. A., and Lebreiro, S. M.: Holocene interdependences of changes in sea surface temperature, productivity, and fluvial inputs in the Iberian continental shelf (Tagus mud patch), Geochemistry, Geophysics, Geosystems, 10, n/a-n/a, 2009.

Please also note the supplement to this comment:

https://www.clim-past-discuss.net/cp-2017-84/cp-2017-84-AC2-supplement.pdf

Figure 1 (Figure 5 of Hernandéz et al., (2015)). NAO, EA and SCAND modes effect on the mean precipitation, temperature and wind speed. 1 month lag time winter months (NDJF avg), and summer months (AMJJA avg).